# Review of Electrochemically Synthesized Resistive Switching Devices: Memory Storage, Neuromorphic Computing, and Sensing Applications

**DOI:** 10.3390/nano13121879

**Published:** 2023-06-17

**Authors:** Somnath S. Kundale, Girish U. Kamble, Pradnya P. Patil, Snehal L. Patil, Kasturi A. Rokade, Atul C. Khot, Kiran A. Nirmal, Rajanish K. Kamat, Kyeong Heon Kim, Ho-Myoung An, Tukaram D. Dongale, Tae Geun Kim

**Affiliations:** 1Computational Electronics and Nanoscience Research Laboratory, School of Nanoscience and Biotechnology, Shivaji University, Kolhapur 416004, India; somnathkundale2025@gmail.com (S.S.K.); girishkamble.snst@gmail.com (G.U.K.); patilpradnya230@gmail.com (P.P.P.); snehalpatil.nanoscience@gmail.com (S.L.P.); curious.kasturi@gmail.com (K.A.R.); 2School of Electrical Engineering, Korea University, Anam-dong, Seoul 02841, Republic of Korea; khotatul94@gmail.com (A.C.K.); kirannirmal57@gmail.com (K.A.N.); 3Department of Electronics, Shivaji University, Kolhapur 416004, India; rkk_eln@unishivaji.ac.in; 4Department of Physics, Dr. Homi Bhabha State University, 15, Madam Cama Road, Mumbai 400032, India; 5Department of Convergence Electronic Engineering, Gyeongsang National University, Jinjudae-ro 501, Jinju 52828, Republic of Korea; kkim@gnu.ac.kr; 6Department of Electronics, Osan University, 45, Cheonghak-ro, Osan-si 18119, Republic of Korea; callname@paran.com

**Keywords:** resistive switching, memristor, electrochemical synthesis, resistive memory, neuromorphic computing, sensor

## Abstract

Resistive-switching-based memory devices meet most of the requirements for use in next-generation information and communication technology applications, including standalone memory devices, neuromorphic hardware, and embedded sensing devices with on-chip storage, due to their low cost, excellent memory retention, compatibility with 3D integration, in-memory computing capabilities, and ease of fabrication. Electrochemical synthesis is the most widespread technique for the fabrication of state-of-the-art memory devices. The present review article summarizes the electrochemical approaches that have been proposed for the fabrication of switching, memristor, and memristive devices for memory storage, neuromorphic computing, and sensing applications, highlighting their various advantages and performance metrics. We also present the challenges and future research directions for this field in the concluding section.

## 1. Introduction

The key to the development of powerful, energy-efficient neuromorphic computer systems lies in the fabrication of electronic devices that can imitate biological synapses. In this context, the efficiency bottlenecks that occur in the conventional computation of neural algorithms have driven the development of novel devices capable of performing in-memory computing. As a result, energy-efficient neuromorphic systems with resistive memory are expected to eventually replace the prevalent Von Neumann computing architecture [1]. Neuromorphic computing emulates the neural networks of the human brain, utilizing neural/synaptic devices that act as processors and memories to process large amounts of data with low energy requirements. The primary motivation for the development of this technology has been the exponential growth of data in recent decades. Data are regarded as the fuel of the new economy and represent a source of wealth in the knowledge era, particularly because they have been generated more rapidly as technology has advanced via multiple sources, including human interactions, electronic media, social networks, machines, and historical archives [2]. The efficient storage and processing of big data have thus taken on greater importance, particularly due to the insights that can be gained from identifying underlying patterns within the data. Therefore, the present review addresses the storage of data and the emergence of neuromorphic computing and sensing applications utilizing resistive switching (RS) devices.

The fabrication of efficient data storage devices for various applications has received significant attention from researchers worldwide [1,3]. One of the noteworthy trends in recent years has been the emergence of memory devices based on memristors, referred to as the fourth fundamental circuit element and pioneered by Prof. Chua [4]. It should be noted that conventional RS devices are also referred to as memristor/memristive devices, meaning these terms can be used interchangeably [5]. In addition to memristive-based memory, solid-state memory devices have become more widespread in the past decade, including those based on resistive random access memory (RRAM) [6], phase-change memory (PCM) [7], magnetoresistive random access memory [8], conductive bridging random access memory [9], and ferroelectric random access memory [10]. Memory devices have also been increasingly employed in both digital computing (as digital gates [11], reconfigurable logic circuits, and crossbar arrays [12]) and analog applications (including neuromorphic computing [13], oscillators [14], amplifiers [15], filters [16], chaotic circuits [17], sensors [18], optical devices [19,20], and thermoelectric devices [21,22]). 

Because of the increasing demands for data storage and information-processing devices for both consumer and industrial applications, recent research has targeted devices with high speed, high memory density, low power consumption, and a small environmental footprint [3]. Accordingly, various physical and chemical deposition techniques have been developed to fabricate RS devices. Commonly used physical deposition techniques include sputtering, atomic layer deposition, e-beam deposition, and pulsed laser deposition. However, given that cost-effectiveness and ease of solution processability are crucial for the fabrication of RS devices, chemical approaches have become more widespread, including hydrothermal methods [23,24], spray deposition [25], SILAR [26], spin coating [27,28], drop-casting [29], microwave [30], screen printing [31,32], doctor blading, and electrochemical methods [33]. In terms of the direction and advancement of RS performance, it is essential to comprehend the impact of synthesis procedures, including the cost. Additionally, synthesis procedures must be defined with regard to their compatibility with complementary metal-oxide semiconductors (CMOSs) and practical deployment. Therefore, it is imperative to conduct a rigorous analysis of the various synthesis techniques present in the literature to enable improvements in device merits.

Electrochemically synthesized RS devices have been reported to exhibit excellent RS performance. Moreover, electrochemical synthesis has several advantages over other chemical synthesis techniques, including a high production rate, low cost, industrial applicability, accurate control over morphology, the possibility of multilayered growth, no requirement for post-deposition treatment, and low operating temperatures [34]. The most widely used electrochemical approaches for the synthesis of thin films are electrodeposition, electrophoretic deposition, and anodization. In these processes, an external electric stress is applied between two electrodes that are immersed in an electrolyte containing ionic species, charged particles, and/or etching agents [35,36,37]. For example, electrodeposition is based on the oxidation and reduction of ionic species due to an external potential/current, with the target material deposited on the working electrode [35]. Electrochemical approaches also allow for variation in the synthesis parameters (such as the applied electric current and deposition time) to induce structural changes in the synthesized material. In addition, the electrolyte concentration, pH, and temperature can be adjusted to modulate the composition of materials and the morphology [38]. This tunability facilitates the fabrication of high-performance RS devices using electrochemical techniques. The primary advantages of electrochemical synthesis are the ability to control the switching layer size using the bottom-up approach, ionic transport, ease of fabrication, and industrial applicability, rendering this technique suitable for state-of-the-art RS devices.

Given the dramatic increase in the volume of data generated due to technological advances and associated scaling issues, the present review focuses on memory, neuromorphic, and sensor applications based on electrochemically fabricated RS devices. The review provides basic information on RS devices, various synthesis techniques, and application domains for researchers. Moreover, this review article is helpful to determine which synthesis technique is beneficial for future research in RS devices. In particular, the present article provides important research directions for electrochemically synthesized RS devices in terms of memory, neuromorphic computing, and sensing applications. Finally, this review recommends future guidelines for electrochemically synthesized high-performance RS devices. The first section of the review provides a general introduction to memory devices, while Section 2 contains a description of the RS effect (and its mechanisms) and a discussion on unipolar and bipolar RS. The details of RS devices and their applications are then discussed in Section 3, while Section 4 focuses on memory storage, neuromorphic and sensing devices, and the fabrication techniques employed for RS devices. A summary of electrochemical techniques is provided in Section 5, including electrodeposition, anodization, and electrophoretic deposition. Subsequently, Section 6 presents a description of electrochemically synthesized RS materials for memory storage applications, including carbon, zinc oxide (ZnO), copper oxide (Cu_2_O), titanium oxide (TiO_2_), and other metal oxides. Section 7 and Section 8 summarize electrochemically synthesized RS materials for neuromorphic computing and sensing applications, respectively. Finally, the future outlook is summarized in Section 9, and the conclusions are presented in Section 10.

## 2. Resistive Switching Effects 

The RS effect is typically observed in metal/insulator or semiconductor/metal devices, in which the insulator or semiconductor material is sandwiched between the top and bottom metal electrodes (Figure 1a) [39]. RS is a unique property of nano- and micromaterials where the electrical resistivity of a material can take on multiple stable states in accordance with the applied voltage [39,40,41,42,43,44]. The applied voltage induces the migration of either oxygen vacancies or metal ions, which forms conducting filaments within the device and tunes the resistance of the switching layer. Figure 1b presents the general steps involved in the RS process [40]. In the electroforming process, a high voltage is applied to the device for a short time to create conductive filaments from oxygen vacancies or metal ions. Most researchers prefer to employ electroforming to initiate the RS process in a device [45], which can lead to both volatile and non-volatile memory effects. For example, Hu et al. reported a Pt/InGaZnO/W device with volatile RS memory without electroforming and stable bipolar RS after electroforming [46]. After the electroforming process, a relatively low SET voltage can put the device into a low-resistance state (LRS) due to the formation of complete conductive filaments, while the subsequent use of a RESET voltage switches the device to a high-resistance state (HRS), in which the conductive filaments are ruptured [47,48,49].

### 2.1. Unipolar and Bipolar Resistive Switching

RS devices can be classified as unipolar or bipolar depending on the polarity of the SET and RESET voltages and their current–voltage (*I–V*) characteristics. As shown in Figure 1c [41], in unipolar RS devices, switching occurs at the same polarity, while a bias of the opposite polarity is required for bipolar RS devices [50]. In unipolar RS devices, the Joule effect is one of the main causes of the dissolution of the conductive filaments during the RESET process [51]; thus, these devices can be switched using voltages with the same polarity. In contrast, in bipolar RS devices, the dissolution of conductive filaments occurs due to the migration of oxygen vacancies or metal ions during the RESET process. It is possible for some RS devices to exhibit both unipolar and bipolar modes [52,53], with the device structure [54,55] and operating setup [56] affecting the switching mode of the device. 

Endurance is an important characteristic of an RS device and is defined as the number of times the device switches between two or more stable resistance states (i.e., SET/RESET transitions) with an appropriate ON/OFF ratio or memory window (Figure 1d) [3]. Endurance can be measured experimentally in a number of ways, such as recording the *I–V* sweep, current-visible pulsed voltage stress (PVS), and current-blind PVS [3,57]. Similarly, data retention is a measure of the long-term stability of the LRS and HRS (Figure 1e) [42]. In non-volatile RS devices, it is crucial to retain data for an extended period. In general, the minimum acceptable data retention time is 10 years [3]. Another important characteristic of RS devices is the switching time (Figure 1f) [43], which is the time required for a device to transition from an LRS to an HRS or vice versa. According to technical requirements, the switching time between SET and RESET should be less than 10 ns per transition in either direction [3]. This high-speed switching ensures the rapid execution of any task.

### 2.2. Resistive Switching Mechanisms 

The RS mechanisms for electrochemical metallization memory (ECM), valance-change memory (VCM), thermochemical memory (TCM), and PCMP are presented in Figure 2 [58,59,60,61]. The following subsections cover the general details of each RS mechanism.

#### 2.2.1. Electrochemical Metallization Memory

Electrochemical redox reactions play a crucial role in ECM devices, with metal oxidation and reduction within the switching layer leading to the formation and rupture of conduction filaments [40]. In particular, when a positive bias is applied to the top electrode (i.e., the anode), metal cations are generated that migrate to the bottom electrode (i.e., the cathode), where they are reduced to form conductive filaments. This switches the device to the SET process (i.e., an LRS). The RESET process (i.e., an HRS) is achieved by Joule heating or by applying a bias of the opposite polarity to the memory device, which oxidizes the reduced cations, leading to their migration back to the anode. Figure 2a presents an ECM device with Cu as the top electrode and highly doped Si as the bottom electrode [22]. A positive bias switches the device to an ON state via the migration of Cu^+^ ions to the bottom electrode, which is reduced to form conduction filaments. The opposite polarity switches the device to an OFF state by moving the Cu^+^ ions back to the Cu electrode [22]. ECM devices offer several advantages, such as lower SET/RESET voltages and lower power consumption, compared to VCM-type devices while also offering rapid switching times (close to a nanosecond scale) and a high ON/OFF ratio due to the complete dissolution of the conduction filaments [62].

**Figure 2 nanomaterials-13-01879-f002:**
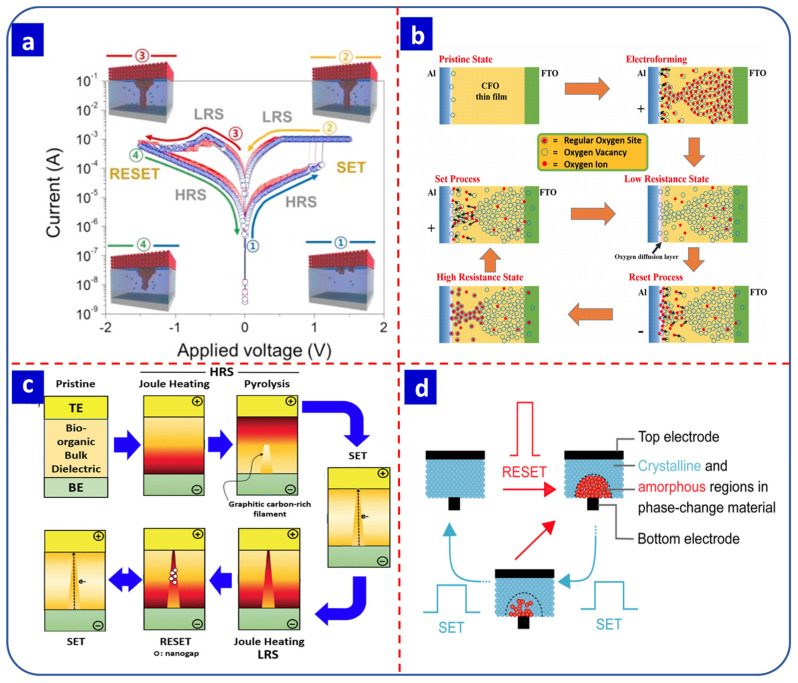
(**a**) Schematic representation of *I–V* sweeping, including the formation and dissolution of a conduction filament in electrochemical metallization memory (ECM). Reprinted with permission from [22]. Copyright © 2023, Elsevier B.V. (**b**) Schematic representation of the electroforming, SET and RESET process in valance-change memory (VCM) due to the migration of oxygen vacancies. Reprinted with permission from [59]. Copyright © 2023, Springer-Nature. (**c**) Schematic representation of the SET and RESET process in thermochemical memory (TCM) [63]. Copyright © 2023, De Gruyter. (**d**) Diagrammatic representation of the amorphous and crystalline state of the cell in phase-change memory (PCM) [61]. Copyright © 2023, Springer-Nature.

#### 2.2.2. Valance-Change Memory

In VCM, oxygen vacancies play a vital role in the RS process. The migration of oxygen vacancies and oxygen ions due to the applied electric field is responsible for the SET and RESET process [62]. RS occurs when a positive bias is applied to the anode, which results in the movement of oxygen vacancies toward the cathode, which acts as an electron acceptor. At the same time, oxygen ions move towards the anode, which switches the device to an LRS by creating oxygen vacancies based on the conductive filaments. On the other hand, the opposite polarity switches the device to an HRS by inserting oxygen ions back into the oxygen sites. The typical VCM-based RS process for an Al/CoFe_2_O_4_/FTO device is presented in Figure 2b [59]. The oxygen vacancy channel is formed during electroforming by applying a high voltage between the top and bottom electrodes. Due to the applied electroforming voltage, oxygen anions migrate towards the positively biased top electrode, leaving behind oxygen vacancies. A sufficient number of oxygen vacancies forms conduction filaments between the two electrodes during the SET process. When a reverse bias is applied, the oxygen anions return and fill the oxygen vacancies near the top electrode, producing an HRS [59]. Compared to ECM cells, VCM cells operate at a higher voltage and thus dissipate more power. However, the retention of a VCM cell is much higher than that of an ECM cell. On the other hand, VCM cells generally have a lower ON/OFF ratio than ECM cells [62].

#### 2.2.3. Thermochemical Memory

TCM is generally the least preferred RS mechanism. Typically, TCM devices exhibit unipolar RS, and the formation and rupture of the conduction filaments are dependent on thermal energy and the Joule effect [58]. In the RESET process, an appropriately high current increases the temperature of the conduction filaments due to the Joule heating effect (Figure 2c) [60,62,63]. Russo et al. reported the simulation analysis of a NiO RS device and the effect of the temperature of conductive filaments on RS properties [60].

#### 2.2.4. Phase-Change Memory

PCM is widely used in non-volatile memory storage devices. In PCM cells, the phase of the switching material (i.e., crystalline or amorphous) is tuned by applying external voltage pulses [64,65]. The resistivity of the amorphous phase is higher than the crystalline phase, allowing for data storage in the form of 1′s and 0′s (Figure 2d) [61]. In the RESET process, applying a large electric pulse to the crystalline material for a short time causes a heating effect that increases the temperature of the switching layer; subsequent rapid cooling leads to an amorphous structure. During the SET process, a medium electric pulse is required to heat the material above the crystallization temperature for a relatively long time to switch the device from an HRS to an LRS [64,66,67]. GeSbTe-based devices employ this phase-change mechanism and exhibit a high endurance of 2 × 10^12^ cycles [68]. Recently, electrochemically fabricated GeSbTe-based PCM devices with excellent non-volatile memory properties have also been reported [69].

## 3. Resistive Switching Devices and Their Applications 

RS devices are widely employed for memory storage [70], neuromorphic [1], and sensing [71] applications (Figure 3). Many metal-oxide-based resistive devices have been studied in recent years due to their unique physical, chemical, and electrical properties that allow them to be employed for memory storage, neuromorphic electronics, logic gate, and sensor applications [72,73,74]. In the following sections, various applications for RS devices are described [75]. 

### 3.1. Memory Storage 

Efficient data storage is a crucial requirement for modern technology, with many memory storage designs having been developed in recent decades. Of these, RRAM devices are the most attractive [73]. A schematic representation of an Al/Ti_3_C_2_T_x_-TiO_2_/Pt RS memory device is presented in Figure 3a, with its *I–V* characteristics shown in Figure 3b and its endurance and retention properties summarized in Figure 3c,d, respectively [28]. 

The technological requirements for RS devices in memory storage applications are summarized in Table 1. In RS memory storage devices, switching the resistance from an HRS to an LRS and vice versa can be used to store data in the form of 0′s and 1′s, respectively [76]. Memory storage devices require low reading and writing voltages, with some RS devices reported to have an operating voltage of 0.5 V with a low power consumption of 0.1 pJ per transition [77]. For practical applications, the endurance of a device should be greater than 10^9^ cycles, with many RS devices achieving up to 10^12^ cycles [57], and its data retention time should approach 10 years, though a recent report for an RS device shows data retention time higher than this [78]. The switching time is also an essential factor for memory storage devices, with a switching time of fewer than 10 ns recently reported for RS-based devices [79]. In addition, the demand for miniaturized memory devices has grown recently, with Govoreanu et al. fabricating a few nanometer-sized RS cells [77]. To date, the most frequently researched materials for memory storage applications have been transition metal oxides [80], metal nitrides [81,82], carbon-based materials [83], polymers [84], and chalcogenides [85,86]. 

### 3.2. Neuromorphic Computing

Neuromorphic computing seeks to emulate brain functions using electronic devices. The human brain is composed of billions of neurons, with synapses controlling their connections. At a basic level, memory and learning are controlled by pre-neurons, synapses, and post-neurons. Typically, a signal is transferred as an electrical spike from a pre-neuron to a post-neuron via a synapse. When the pre-neuron receives the signal, transmitters are generated in the vesicles, and depolarization of the pre-neuron opens the channel, through which the transmitters reach a receptor in the post-neuron by passing through a 20–40 nm long synaptic cleft [1]. In the case of artificial synapses, the properties of a biological synapse are emulated using metal/insulator/metal RS devices, with the top and bottom electrodes acting as the pre-neuron and post-neuron, respectively, and the switching layer or insulator acting as the synapse. For the more accurate mimicking of synaptic properties, analog RS-based devices are generally used, with metal oxides [1], perovskites [87], polymers [88], organic materials [89], and 2D nanomaterials [90] being widely used for RS-based neuromorphic computing applications. A schematic representation of biological and electrical synapses based on RS devices is presented in Figure 3e [86].

**Table 1 nanomaterials-13-01879-t001:** The technology requirements of RS devices for memory storage, neuromorphic computing, and sensor applications.

	Requirements		Refs.
Memory storage	Operating voltage	<1 V	[3,76,77,78,79]
	Type of switching	Digital, analog	
	Switching layer thickness	50 nm	
	Power consumption	~10 pJ/transition	
	Switching time	<10 ns/transition	
	Endurance	>10^9^ cycles	
	Data retention	>10 years (85 °C)	
	MIM cell size	~500 nm^2^	
	3D stacking	Yes	
Neuromorphic computing	Operating voltage	<1 V	[1,2,91]
	Type of switching	Analog	
	Switching layer thickness	50 nm	
	Power consumption	~10 pJ/transition	
	Switching time	<10 ns/transition	
	Endurance	>10^9^ cycles	
	Data retention	>10 years	
	MIM cell size	~500 nm^2^	
	Potentiation and depression (P&D)	Yes	
	Spike-timing-dependent plasticity (STDP)	Yes	
	Spike-rate-dependent plasticity (SRDP)	Yes	
Sensor	Operating voltage	<1 V	[3,75]
	Type of switching	Digital, analog	
	Switching layer thickness	50 nm	
	Power consumption	~10 pJ/transition	
	Switching time	<10 ns/transition	
	Endurance	>10^9^ cycles	
	Data retention	>10 years (85 °C)	
	MIM cell size	~500 nm^2^	
	Response time	Low as possible	

Synaptic plasticity is an essential property of the human brain that is associated with modifications in the synaptic weight, which is the strength between two neurons. Spike-timing-dependent plasticity (STDP) and spike-rate-dependent plasticity (SRDP) are the two main variants of synaptic plasticity [1]. Positive and negative values for the synaptic weight represent potentiation and depression, respectively, which can further be classified into short-term potentiation (STP), long-term potentiation (LTP), short-term depression (STD), and long-term depression (LTD) [91], all of which facilitate information storage and learning. STD involves a temporary rise in the synaptic weight followed by an abrupt decline to the original state. In LTD, a permanent change in the synaptic weight occurs following repeated input pulses [91]. A temporal decay in the negative synaptic value indicates STD, while LTD is identified by a permanently low negative synaptic weight [1,91]. The potentiation–depression and antisymmetric Hebbian learning rules of a Ag/a-BN/Pt memristor device are displayed in Figure 3f,g, respectively [82].

The fabrication of electronic devices to mimic synaptic properties (such as STD and LTD) remains challenging. Recently, many electronic devices have been reported that mimic basic and advanced brain functionalities, including RS-based memory devices [92], transistors [93], and ferroelectric junctions [94]. However, the current electronic synaptic devices generally have high power/energy consumption, unstable potentiation and depression dynamics, and unstable conduction states. In addition, the non-linearity and fewer resistive states of artificial synaptic devices have become bottlenecks, affecting the realization of high-performance neuromorphic computing systems. Therefore, further research is required to overcome these challenges and produce efficient artificial synapses. The technological requirements for RS devices in neuromorphic computing applications are summarized in Table 1.

### 3.3. Sensor Applications

RS devices can be employed in both sensors and detectors [75,95]. Most resistance-based sensing devices operate based on a change in resistance induced by a change in the properties of the constituent materials. In RS-based electrochemical sensors, the oxidation and reduction of constituent materials in accordance with the applied voltage and presence of the analyte are monitored [96]. In this context, the change in the resistance from an LRS to an HRS (or vice versa), switching voltage, hysteresis area, and memory window are important physical parameters for RS-based sensors. In particular, when an RS device comes into contact with a target analyte, one or more of these physical parameters will generally change, and monitoring this change allows the device to be employed as a sensor. A representative RS-based sensor is presented in Figure 3h–j [75]. In this case, the change in the current over time and in response to the concentration of the analyte when applying an appropriate voltage shows that resistive materials can respond to the analyte and switch to a different resistive state [75]. The analyte can thus be analyzed quantitatively or qualitatively based on the gap in the voltage or the change in RS properties [49]. The technological requirements for RS devices in sensor applications are summarized in Table 1.

### 3.4. Other Applications

Other applications that can employ RS memory characteristics include thermoelectric [21,22] and optical devices [97]. These are applications for solving the trade-off between conductivity and transmittance [98,99] (Figure 3k–m).

**Figure 3 nanomaterials-13-01879-f003:**
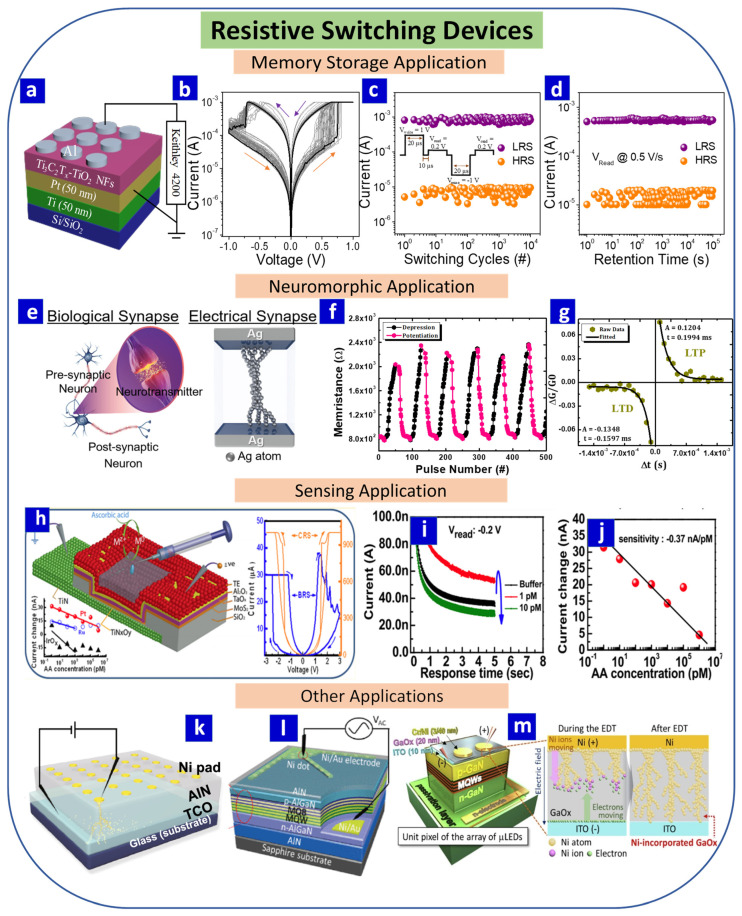
(**a**) Schematic illustration of an Al/Ti_3_C_2_T_x_-TiO_2_/Pt RS device for memory storage applications. (**b**) *I–V* characteristics of the device. (**c**) Endurance and (**d**) retention properties of the device. Reprinted with permission from [28]. Copyright © 2023, published by Elsevier. (**e**) A schematic representation of biological and electrical synapses. Reprinted with permission from [86]. Copyright © 2023, published by Elsevier B.V. (**f**,**g**) Neuromorphic properties, potentiation, and depression behaviors of a Ag/a-BN/Pt memristor device. Reprinted with permission from [82]. Copyright © 2023, American Chemical Society. (**h**) Schematic representation of an IrO_x_/Al_2_O_3_/TaOx/MoS_2_/TiN device for ascorbic acid (AA) detection. (**i**) Current vs. response time curves for this device at buffer and different AA concentrations. (**j**) Current change vs. AA concentration (1 pM to 1 μM) measured using this device. Reprinted with permission from [75]. Copyright © 2023, American Chemical Society. (**k**) Schematic of electric-field-driven metal implantation for electrode fabrication. Reprinted with permission from [97]. Copyright © 2023, Springer Nature. (**l**) Schematic illustration of an AlGaN-based DUV light-emitting diode. Reprinted with permission from [98]. Copyright © 2023, IEEE. (**m**) Electric-field-induced doping treatment process and thermal stability of the current paths formed in a GaOx electrode. Reprinted with permission from [99]. Copyright © 2023, Elsevier B.V.

## 4. Fabrication Techniques for Resistive Switching Devices 

The method employed to fabricate a device will affect its RS properties [100,101,102], with the same material potentially demonstrating different RS behavior when synthesized using different approaches [103,104]. Thus, various factors such as industrial practicality, cost-effectiveness, and RS performance need to be considered when fabricating an RS device. Techniques such as sputtering [105,106,107], lithography [108], spin coating [109], spray pyrolysis [110], and electrodeposition [104] have previously been reported for the synthesis of the active switching layer in RS devices, while hydrothermal [111], dip coating [112], SILAR [26], electrospinning [113], and many other solution-processable techniques [114,115] have also been developed for the same purpose. For the top and bottom electrodes, physical deposition techniques are utilized; thus, this section focuses on the options for the fabrication of the active layer.

The advantages and disadvantages of switching layer synthesis techniques are summarized in Table 2 [116,117,118,119,120]. Physical fabrication techniques such as sputtering, atomic layer deposition, and pulsed layer deposition are precise and suitable for industrial applications. However, they are costly and require many steps to fabricate cost-effective RS devices, while their high operating temperatures, requirement for a vacuum, and inability to control the morphology are also limitations. For example, as presented in Figure 4a, the fabrication process for a Pt/HfO_2_/Ti RRAM device involves two-step lithography, etching, and deposition processes that are quite complex [100]. Similarly, chemical vapor deposition requires an inert atmosphere and high temperatures. In contrast, spin coating, hydrothermal fabrication, and spray pyrolysis represent cost-effective and easy fabrication approaches. However, spin-coated and hydrothermally synthesized thin films have lower adhesion and durability than physical vapor deposition and are thus not suitable for large-scale deposition. Figure 4b presents the process for the preparation of a spin-coated Cu/HfO_2_/p++ Si device [27]. The high synthesis temperature required for these techniques is also a disadvantage. Figure 4c presents the steps involved in the fabrication of a Ag/CuO/SiO_2_/p-Si RS device, which involves the deposition of a CuO thin film on a SiO_2_ substrate using spray pyrolysis and the deposition of the top electrode using e-beam evaporation [101]. 

Figure 4d presents an anodized aluminum oxide (AAO) template-based electrochemical approach for RS device fabrication [102]. Electrochemical synthesis techniques generally involve a simple fabrication process and are industrially feasible, inexpensive, environmentally friendly, do not require high temperatures, and produce consistent growth over a large surface area. Moreover, they allow precise control of the film thickness and achieve high switching quality through adjustment of the synthesis parameters. In addition, the adherence and durability of electrochemically synthesized thin films are significantly higher than those with other solution-processable techniques. Electrochemical approaches can also be used to fabricate the entire metal–insulator–metal device, while template-assisted deposition and nanostructured devices can be achieved with electrochemical synthesis. Some reports have also suggested that electrochemically synthesized devices have superior properties compared with sol–gel and spin-coated devices [103,104]. The advantages and disadvantages of the synthesis methods are summarized in Table 2 [116,117,118,119,120]. The thickness of the switching layer can also affect the RS properties of the end device, and electrochemical synthesis allows the deposition of a very thin switching layer with controlled morphology. For example, the minimum thickness of a switching layer achieved using electrodeposition is 15–20 nm [121]. 

Various criteria should be considered when comparing the RS performance of memory devices made using diverse fabrication technologies, such as the thicknesses of the switching and top and bottom electrode layers, the choice of top and bottom electrodes, and the synthesis methods because they all have an influence on the RS characteristics. We analyzed the RS performance of various copper oxide-based devices by considering the previously mentioned factors while focusing on industrially applicable techniques that are cost-effective with a simple synthesis process, as summarized in Table 2. From the table, it is evident that electrochemically synthesized copper oxide-based devices are switched at a low voltage (0.7 V), similar to RS devices fabricated using a sputtering process (V_SET_: 0.7 V and V_RESET_: −0.5 V). In addition, devices made using these techniques exhibited endurance greater than 10^3^ cycles and retention up to 10^3^ s, while devices fabricated using other methods achieved relatively poor RS performance [104,105,106,107,108,109,110,111,112,113,114,115,116,117,118,119,120]. These results indicate that the electrochemical method can provide excellent RS performance, similar to sputtering and other physical synthesis procedures.

## 5. Electrochemical Methods for Resistive Switching Devices

### 5.1. Electrodeposition 

Electrochemical synthesis is an elegant and versatile technique that allows the controlled deposition of an electrolyte onto conducting material with the application of an electric current or voltage across two electrodes [122]. Electrochemical synthesis can be classified as cathodic and anodic electrodeposition [123]. In cathodic electrodeposition, charged species, clusters, and nanoparticles are deposited on the cathode. Depending on the deposited charged species, charged particles, and applied power, electrodeposition can be classified as electroplating, electrophoretic, or electroless deposition. In electroplating, metal cations in the electrolyte move towards the cathode due to an external electric field and are reduced on the surface of the cathode (Figure 5a). Most semiconductors, chalcogenides, and oxide-based thin films can be fabricated using this technique. 

In electrophoretic deposition, similar reaction mechanisms can occur, but it does not involve any chemical reaction, i.e., the reduction of ionic species. Instead, the driving force for this technique depends on the charge and electrophoretic mobility of the particles involved. The electrolyte is replaced by a suspension of charged particles in an organic solvent [123]. This technique can be used to deposit charged particles, nanomaterials [124,125], and carbon-based materials [126]. Figure 5b presents the general mechanisms for electrophoretic deposition. Dispersed charged particles in the organic solvent are attracted or deposited on the oppositely charged conducting electrode following the application of an external DC voltage/current. Electrophoretic deposition can be both cathodic and anodic [127]. 

In electroless deposition, deposition occurs without any external electric supply. This is an autocatalytic process in which redox reactions occur due to reducing agents that act as a source of electrons (Figure 5c) [123,128]. The electrode is dipped in an electrolyte containing reducing agents and metal precursors. The autocatalytic redox reactions deposit metal atoms on the electrode by removing the metal atoms on the electrode surface. This deposition technique provides a simple and cost-effective path for the decoration of metals such as Sn, Au, and Ni or nanoparticles on metal-oxide thin films [129,130] and is highly selective. 

Electrochemically prepared thin films can be deposited in a potentiostatic or galvanostatic manner, in which the current density varies with a constant voltage and the electrode potential varies with a constant current, respectively. Chronoamperometry, chronopotentiometry, and voltammetric deposition are also common thin-film synthesis techniques [104]. In electrochemical fabrication, several parameters should be considered for the uniform deposition of thin films, including the electrolyte concentration, applied electric field, pH, deposition time, deposition temperature, and distance between the two electrodes [131,132]. The thickness of the deposited film is controlled by tuning the deposition time and the applied electric field, while pH and the deposition temperature are responsible for compositional changes in the deposited materials [38]. In addition, the distance between the two electrodes can determine the structure and morphology of the thin film. 

**Figure 5 nanomaterials-13-01879-f005:**
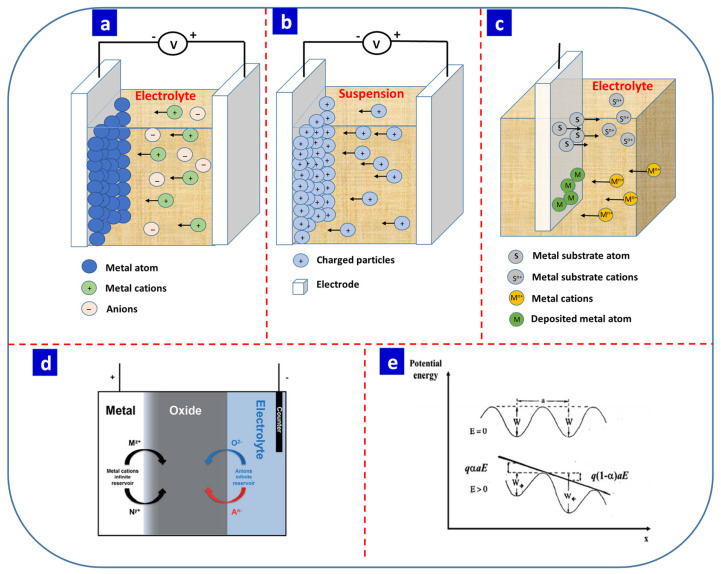
(**a**) Schematic representation of electrodeposition using a two-electrode system. (**b**) Schematic representation of electrophoretic deposition and (**c**) electroless deposition. (**d**) Schematic representation of anodization. (**e**) Influence of the electric field strength on the activation energy for the migration of ions during anodization. Reprinted with permission from [132]. Copyright © 2023, Royal Society of Chemistry.

### 5.2. Anodization

Anodization is a widely used technique for the preparation of high-quality thin films for the fabrication of electronic devices due to its large-scale fabrication capabilities and simplicity [131,132,133]. In the anodization process, the metal anode is electrochemically oxidized. Figure 5d presents the metal anode as a reservoir of infinite metal cations (M^z+^/N^y+^) with an electrolyte reservoir of infinite anions (O^2−^/A^n−^) [132]. The applied electrical potential oxidizes the cations on the surface of the anode, which forms a thin oxide film on the surface of the anode. The thickness of the oxide layer is directly proportional to the applied electric field, which triggers the ionic conductivity of the oxide on the surface of the metal. Three theories have been proposed to explain the rate-determining steps: Verwey’s theory, which proposes that ion transfer within the oxide film is responsible for the metal-oxide thin film [134], Mott-Cabrera’s theory, which posits that ion transfer across the metal–oxide interface and determines the thickness of the film [135], and Dewald’s theory, which highlights the influence of ion transfer across the oxide–electrolyte interface [136]. The high field mechanism explains the exponential dependency of the current density (*i*) on the electric field strength (*E*), as given in Equation (1) [132].
(1)$$\;i\;=N\nu q\exp\;\left(-\frac{W-\alpha aqE}{k_{B}T}\right)$$
where *N* is the surface density of mobile ions, *v* is the vibrational frequency of mobile ions, *q* is the charge associated with the mobile ions, *W* is the activation energy, α*a* is the activation distance, *k_B_* is the Boltzmann constant, and *T* is the absolute temperature. The applied electric potential equations can be derived by assuming that the ions gain sufficient potential energy via thermal agitation. Figure 5e presents the potential energy of ions in relation to the distance with and without an electric field. The former is assumed to reduce the height of the barrier from *W* to (*W − qE*)*,* while the latter increases the barrier for ions moving against the field from *W* to (*W* + *qE*) [132].

A strong electric field during anodization can cause the metal cations to migrate towards the metal-oxide–electrolyte interface, while O^2−^ or anions are inserted into the metal oxide and reduced at the metal–metal-oxide interface. The oxidation of metal cations occurs within the metal oxide, at the metal–metal-oxide interface, or at the metal-oxide–electrolyte interface. Anodization parameters such as the applied voltage and anodization time influence the thickness and crystallinity of the resulting oxide film, while the choice of the electrolyte affects the composition of the oxide layer and the pH of the solution and etching agents determine the morphology of the oxide thin film.

## 6. Electrodeposited Resistive Switching Materials for Memory Storage Applications

Minimal research has been conducted on electrochemically synthesized thin films for memory storage applications. However, the most frequently researched electrodeposited materials for RS device fabrication are carbon-based materials, metal oxides, and chalcogenides. The following subsections discuss the use of these materials.

### 6.1. Carbon-Based Resistive Switching Devices

There are very few reports available on carbon-based RS devices [137]. In recent years, graphene oxide [138,139], diamond-like carbon [140], carbon quantum dots [141], single-walled carbon nanotube (SWCNT)@TiO_2_ core–shell wires [142], graphene oxide-TiO_2_ [143], self-generated amorphous carbon [144], graphene-integrated devices [145], and carbon nanotubes have been explored for use in RS memory applications [146], but very few of these have been electrochemically synthesized. Carbon materials have numerous advantageous properties related to their morphology, defects, percentage of oxygen groups, and sp^2^ and sp^3^ carbon atoms, and these properties can be utilized to tune the RS effect of a device [35,137]. 

Russo et al. reported a new electrochemically deposited carbon material for RS devices with free bipolar RS properties [35]. They generated carbon nanowalls (CNWs) via electrophoretic deposition from arc-discharged polyyne electrolytes and demonstrated that the defects in and the orientation of the CNWs affected their electrical resistivity, with the presence of oxygen vacancies and a perpendicular orientation leading to an LRS in an Al/CNW/FTO device. Additionally, trapping and de-trapping electrons in oxygen vacancies with a positive and negative bias switches the device between an LRS and an HRS, respectively [35]. In 2019, Russo et al. reported the electrophoretic deposition of carbon material for memory application [137]. They employed a two-step synthesis approach that combined the electrophoretic deposition of the carbon material with subsequent electrochemical oxidation to produce an active switching layer (Figure 6a) [137]. They reported that the content of oxygen and carbonyl groups (C=O) in the electrochemically oxidized carbon material controlled the RS properties. The anodically oxidized carbon material exhibited three-level memory behavior and a higher ON/OFF ratio than non-oxidized carbon materials. Figure 6b,c present the cyclic stability and retention of the Al/oxidized carbon structures@FTO device [137], with the degree of oxidation well controlled by the anodic oxidation parameters. The oxidized carbon material exhibited RS behavior within a ±2 V range. While a higher percentage of carbonyl groups (C=O) did not provide a higher ON/OFF ratio, the amount of sp3 and sp2 carbon atoms and oxygen-containing carbon groups did play a key role in the Al/OC@FTO device [137]. Liu and co-workers also reported that the electrophoretic deposition of graphene oxide could potentially be used in memristive devices [147]. This is because the oxygenation levels of graphene oxide can affect the properties that are tuned using electrophoretic parameters. Their fabricated Al/GO/ITO device had an ON/OFF ratio of more than 10 and a retention time of 100 s [147]. Table 3 summarizes the electrochemically prepared carbon-based materials for memory applications. This summary highlights a considerable research gap in terms of electrochemically synthesized carbon materials for RS-based memory devices.

### 6.2. ZnO-Based Resistive Switching Devices

In recent years, transition metal oxides have attracted significant attraction due to their CMOS compatibility and ease of synthesis [148]. ZnO is the most common metal oxide used in electrochemical synthesis. As with carbon materials, the orientation and morphology of ZnO can affect the RS properties. Jing et al. synthesized ZnO-based RS devices, and Figure 7a presents the structure of their electrodeposited ZnO thin-film-based device [36]. They produced nanoflake, agglomerated nanoparticle, and nanorod morphologies by tuning the deposition temperature and found that the Au/ZnO nanorod/FTO device had strong RS properties, while the ZnO nanoflakes and nanoparticles did not (Figure 7b). The ZnO nanorod-based device exhibited an endurance of 2000 cycles and retained data for up to 1800 s (Figure 7c,d, respectively) [36]. In another study, Yusoff et al. synthesized ZnO nanowires using two-step electrodeposition and thermal oxidation. They varied the deposition time to investigate the effect of the nanowire size, the thickness of the film, and the RS properties of the device [149]. 

A cyclic voltammetric deposition is an essential form of electrochemical deposition. In this technique, an applied voltage is swept between specific voltage ranges, producing a thin film with superior surface coverage compared with conventional electrodeposition [104]. Sun et al. fabricated a Au/ZnO/AZO structure in which ZnO thin film was cathodically electrodeposited in a zinc nitrate solution [150]. The electrochemical deposition was carried out potentiostatically in a conventional three-electrode cell at −0.9 V with a constant temperature of 353 K and a deposition time of 1000 s. The RS properties were tuned by simply adding nitric acid to the electrolyte. They reported that the reaction of nitric acid with OH- ions in the electrolyte decreased oxygen availability, leading to more oxygen vacancies and n-type conductivity in ZnO. The ON/OFF ratio significantly decreased with an increase in the nitric acid concentration. A low concentration of nitric acid led to few oxygen vacancies while a high concentration did not induce the RS effect. When the carrier concentration was higher than 10^19^ cm^–3^, the materials did not exhibit any RS properties [150]. 

Fuzi et al. also fabricated a flexible Au/ZnO/ITO/PET RS device using electrodeposition. They reported that deposition time is an essential parameter when tuning the RS properties [148]. Figure 7e presents the semi-logarithmic *I–V* curves for electrochemically prepared ZnO thin films deposited for 15, 30, 60, and 120 s. Above 60 s, the device did not exhibit any notable *I–V* characteristics, while the R_OFF_/R_ON_ ratio decreased with increasing deposition time (Figure 7f). The combination of electrochemical with other techniques assisted in the oxidization of Zn, leading to stronger RS properties than electrochemically deposited ZnO. In particular, thermal oxidation is commonly used in conjunction with electrochemical synthesis. For example, Quintana et al. synthesized faceted and spongy single-crystal ZnO using a combination of electrodeposition and hydrothermal techniques [151]. The faceted and spongy Zn rods were synthesized galvanostatically and potentiostatically, respectively, followed by hydrothermal treatment in distilled water to produce ZnO rods. The end device exhibited good *I–V* and retention properties. 

Many research groups have synthesized composite, doped, bilayer, and decorated semiconductor materials to improve RS properties. Recently, Younis et al. synthesized a Ti-doped ZnO thin film from Zn(NO_3_)_2_·6H_2_O via electrodeposition on ITO [152]. The Au/Ti@-ZnO/ITO device exhibited electroforming-free and current compliance-free memory effects and typical bipolar RS properties with good SET (2.9 V) and RESET (−2.7 V) voltages. The device was stable for up to 200 cycles and retained data for up to 2000 s. The memory margin of the device with a 2% Ti@ZnO structure was much higher than that of the control device (pure ZnO) [152]. 

Bilayered and trilayer devices have also been shown to have tunable RS properties. For example, Fauzi et al. electrodeposited ZnO on a copper substrate followed by thermal oxidation to produce a Au/ZnO-Cu_2_O-CuO/Cu RS device. Figure 8a,b present the electrodeposition setup and device structure, respectively [153]. A trilayer structure was observed due to thermal oxidation. The active layer thickness (ZnO-Cu_2_O-CuO) was approximately 81 nm. The authors reported that the HRS/LRS ratio was directly proportional to the thickness of the thin film (Figure 8c,d); as the thickness increased with longer deposition times, the difference between the resistance of the LRS and HRS and the HRS/LRS ratio tended to decrease. In 2018, Fauzi et al. deposited Ti and Zn on a Cu substrate using dilute electrodeposition and subsequently oxidized it thermally at 600 °C. They observed that the Au/TiO_2_-Cu_2_O-CuO/Cu device had a higher HRS/LRS difference than Au/ZnO-Cu_2_O-CuO/Cu. They suggested that metal ion formation was responsible for the RS effect in their device [154]. 

The decoration of ZnO with quantum dots can also enhance RS properties. Younis et al. electrochemically synthesized ZnO nanorods decorated with CeO_2_ quantum dots (∼5 nm; Figure 8e) [155]. The average diameter and length of the ZnO nanorods were ∼20 and ∼200 nm, respectively. The fabricated Au/CeO_2_−ZnO/FTO device exhibited stronger memristive properties than the Au/ZnO/FTO device (Figure 8f). In terms of the temperature dependence of the RS characteristics, the CeO_2_−ZnO nanocomposite showed no change in the memory window. However, the memory window of the ZnO nanorod-based device tended to decrease as the temperature increased. Figure 8g,h depict the endurance and retention properties of the Au/CeO_2_−ZnO/FTO device. Their results suggested that the decoration of CeO_2_ quantum dots improves the RS characteristics of electrochemically synthesized ZnO nanorods [155]. The electrochemical approach is also useful for tuning the charge transport within a device. Ocampo et al. fabricated nanostructured porous Si-metal-oxide (ZnO and VO_2_) composites, with the porous Si-ZnO device exhibiting better memristive properties than the Si-VO_2_-based RS device due to the higher porosity of the ZnO, which facilitated charge transport within the active switching layer [156]. 

Table 4 summarizes the electrochemically synthesized ZnO-based RS devices that have been reported to date. The electrodeposition technique has frequently been used for material synthesis, but further research is warranted on the synthesis of anodization-based ZnO RS materials. It is important to note that electrodeposited ZnO RS devices require higher switching voltages than physically deposited RS devices, while their memory window, endurance, and retention properties are also lower. Given this, research should focus on improving the non-volatile memory performance of electrochemically synthesized ZnO RS devices. 

### 6.3. TiO_2_-Based Resistive Switching Devices

Memory storage devices based on transition metal oxides have been well studied in recent years. In particular, TiO_2_ has been the most widely researched material for RS memory applications. In 2010, Miller et al. synthesized titanium dioxide using anodization for memory storage applications and investigated the memristive properties of both annealed and nonannealed devices. The nonannealed anodized TiO_2_ exhibited a pinched hysteresis loop, while the annealed films did not demonstrate any significant RS properties. There is no need for the post-deposition annealing of anodized films due to the inherent presence of oxygen vacancies at the Ti/TiO_2_ interface [37]. There have been very few reports available on the fabrication of TiO_2_ nanotubes using electrodeposition for memristor applications, but it is clear that electrodeposition can produce TiO_2_ nanotubes for RS applications [157]. Power consumption is an important aspect of RS devices that depends on the operating voltage. Therefore, there is a need to fabricate devices with a lower switching voltage. In this respect, anodically prepared Au/TiO_2_/Ti devices can be operated within ±0.3 V [158]. 

The anodization of Ti foil in different electrolytes leads to the formation of TiO_2_ nanotubes with different shapes, leading to devices with different RS properties. For example, Nirmal et al. synthesized highly ordered TiO_2_ nanotubes via the anodization of Ti foil, with the proposed growth mechanisms presented in Figure 9a. They deposited Al as the top electrode via sputtering, and the resulting Al/TiO_2_/Ti device (Figure 9b) exhibited RS within ±2 V (Figure 9c). The endurance and retention of the fabricated device are displayed in Figure 9d,e, respectively. They proposed a filamentary-type RS model (Figure 9f) and mimicked different STDP rules utilizing the analog behavior of the proposed memristive device (Figure 9g,h) [133]. Similarly, Yoo et al. synthesized highly ordered TiO_2_ nanotubes via the anodization of Ti foil in an HF/H_3_PO_4_ electrolyte. The TiO_2_ nanotubes had a V-shaped morphology and a short aspect ratio, which was suitable for Pt coverage. The Pt/TiO_2_/Ti device exhibited RS within ±1 V [159]. 

Similarly, Yin et al. fabricated a Pt/TiO_2_/Ti device in which TiO_2_ was deposited via anodization in an H_2_SO_4_ electrolyte. Interestingly, four stable resistance states were observed in the prepared device. The eightwise and counter-eightwise *I–V* curves indicated the coexistence of ion- and electron-dominated RS mechanisms [160]. The choice of the top electrode in an anodic TiO_2_-based RS device influences its RS properties. Given this, Yu et al. reported an anodic TiO_2_-based RS device with Ag and Au as the top electrodes. The Au-based device did not exhibit any notable *I–V* characteristics; however, with the Ag top electrode, improved *I–V* characteristics were observed. They also reported that anodizing for only 5 min produced a TiO_2_ device with more suitable properties than the anodization times of 1, 3, and 10 min. Thus, the deposition time is also a very sensitive parameter for electrochemically synthesized RS devices [161].

Tao et al. fabricated a Cu/TiO_2_/Ti nanopore array membrane device using two-step anodization [162]. First, Ti foil was anodized in an NH_4_F-based ethylene glycol electrolyte and washed in an ultrasonicator. It was then anodized in the same electrolyte under an ice bath at 20 V, 40 V, and 60 V. The devices anodized at 20 V and 40 V did not require electroforming steps. RS occurred within a ±1 V window, and the R_HRS_/R_LRS_ ratio increased as the anodization voltage increased. Generally, electrochemical metallization was observed in the device with a Cu top electrode, but the Cu/TiO_2_/Ti device exhibited Ohmic conduction. This suggested that the oxygen vacancies led to the formation of conduction filaments in the TiO_2_ during the SET process. The device was composed of Cu nanocones and Ti nanopores, with a highly concentrated electric field between the two, which enabled the migration of oxygen vacancies [162]. Chen et al. also fabricated core–shell copper nanowire-TiO_2_ nanotubes (TNTs) via anodization and electrodeposition [163]. The TNTs were synthesized using two-step anodization, while the Cu nanowires were grown on the TNTs using electrodeposition. The RS properties were investigated for both Au/TNTAs/Ti and Au/Cu nanowire-TNTAs/Ti devices, with the Cu nanowire-TNTA-based device exhibiting both analog and digital RS properties. The device was stable for 10^4^ cycles and was able to retain data for 10^4^ s [163]. 

Table 5 summarizes the previously reported electrochemically prepared TiO_2_-based RS devices [37,157,158,159,160,161,162,163,164,165,166]. Unlike ZnO, anodization was frequently used for the synthesis of TiO_2_ for RS memory applications. All of the electrochemically prepared TiO_2_ RS devices have demonstrated suitable switching voltages. However, improvements in the memory window, endurance, and retention properties of electrochemically prepared TiO_2_ RS devices are required. 

### 6.4. Cu_2_O-Based Resistive Switching Devices

Cu_2_O has been frequently employed for various applications, including memory [167,168,169], photocatalysis [170], solar cells [171], supercapacitors [172], and gas sensing [173]. However, Cu_2_O has not been well explored for RS applications. Cu_2_O has the potential to be compatible with integrated circuit fabrication processes. Yazdanparast et al. reported unipolar RS properties for electrodeposited Cu_2_O on a Au-coated glass substrate in a highly alkaline tartrate bath. The RS of their device was attributed to the formation and rupture of Cu filaments, with an initial resistance of 6.5 × 10^6^ Ω. The LRS of the device demonstrated a linear decrease in resistance with decreasing temperature, suggesting the formation of Cu filaments. The calculated diameter (70 nm) and resistance temperature coefficient (1.57 × 10^−3^ K^−1^) in an LRS confirmed the formation of Cu filaments, which were nearly equal to Cu nanowires. The authors also calculated the temperature (798 ± 55 K) of the Cu filaments during the RESET process, which caused the rupture of the filaments due to Joule heating [174]. In another study, the same research group reported the effect of various electrodeposited parameters on RS properties. Electrodeposition voltage can be used to tune the size of the electrodeposited microstructure and RS properties, and their study found that decreasing the applied negative voltage increased the grain size of Cu_2_O. Additionally, the grain size increased with an increase in temperature. They observed that the forming voltage decreased with an increase in the electrodeposited negative potential and an increase in temperature. Furthermore, the SET/RESET voltage was dependent on the grain size of the material. In addition, the LRS current did not depend on the grain size or the surface area of the top electrode, which suggested that filamentary conduction is the dominant mechanism in Au/Cu_2_O/Au-Pd devices [175].

Fabrication of high-density memory storage devices is a major challenge that can be achieved by scaling down the size of the device and developing cross-point array structures, which require precise, large-scale, and selective area deposition methods. In this respect, electrochemical deposition can be used to precisely fabricate active switching layers. Recently, Han et al. fabricated a memory device by electrodepositing Cu_2_O (70 nm) within the holes of a SiO_2_ substrate on a Pt bottom electrode. The top Pt electrode was deposited using an e-beam evaporator (Figure 10a) [167]. The size of the cell is presented in Figure 10b [167]. The device exhibited bipolar RS within ±1 V and a forming voltage within ±1 V, representing a low power consumption (Figure 10c) [167]. The fabricated device also demonstrated good retention (Figure 10d) and endurance (Figure 10e) [167]. In another study, the same research group fabricated Cu/CuO_x_/Cu devices using the electrodeposition of Cu, CuO_x_, and Cu on an AAO template (Figure 10f) [168]. The active layer thickness and pore diameter of the AAO template influenced the SET/RESET voltage of the devices. The device structure and *I–V* characteristics of the electrochemically deposited Cu/CuO_x_/Cu devices are presented in Figure 10g [168]. The RS mechanisms of these devices depended on the top and bottom electrode material because the ionization energy of Cu is lower than that of Cu_2_O. The authors suggested that, when a positive bias was applied to the bottom electrode, the Cu induced the diffusion of Cu^+^ ions into the Cu_2_O layer. Furthermore, the Cu ions drifted from the top electrode to the bottom electrode and formed Cu conduction filaments in the switching layer. The filaments were ruptured by applying a negative bias to the bottom Cu electrode (Figure 10h) [168]. Patil et al. also fabricated Al/Cu_2_O/FTO memristive devices based on an electrochemical approach, showing that the space charge-limited current (SCLC) mechanism was responsible for conduction, while RS occurred due to the filamentary effect [176]. 

Flexible memory storage devices are the future of ubiquitous computing systems. To fabricate a flexible memory device, a flexible substrate is employed to support the entire device. Recently, Park et al. electrodeposited Ni/CuO_x_/Ni on a flexible polyethylene terephthalate (PET) substrate, producing a device with reliable RS and endurance. The fabricated device exhibited strong performance under flat, tensile, and compression stress [177]. Kim et al. also fabricated a self-compliance-based bilayered memory device with a Pt/CuO_x_/CuO_x_/Pt structure. The electrical properties of the CuO_x_ layers were tuned by controlling the pH of the electrolyte. In their device, one CuO_x_ layer acted as an internal resistor while the other operated as an RS layer. Due to the internal resistor, the LRS current was controlled and reproducible RS properties were observed without current compliance [178]. Electrochemical approaches can also be extended to the fabrication of metal sulfide-based RS devices. For example, Yan and co-workers fabricated Ag/Cu_2_S/Ag and Ag/Cu_2_S/Cu memristive devices and recorded good RS performance [179]. 

Table 6 presents previously reported electrochemically synthesized Cu_2_O-based RS devices. The RS performance of these devices is not yet satisfactory. More attention should be focused on improving their memory window, endurance, and retention properties. Furthermore, electrodeposition has frequently been employed to synthesize Cu_2_O-based RS devices; thus, other electrochemical approaches should be researched to achieve high-performance RS memory devices. The choice of the electrolyte in electrochemical synthesis affects the morphology and composition of the active layer material, which impacts the RS properties of the end device. For Cu_2_O-based RS devices, CuSO_4-_based electrolytes have been preferred in combination with acid and NaOH. Lactic acid, sulfuric acid, and tartaric acid have been the most preferred choices of researchers for the electrochemical synthesis of Cu_2_O-based RS devices. Therefore, RS properties could be improved further by testing other electrolytes during the synthesis of Cu_2_O.

### 6.5. Other Metal-Oxide- and Composite-Based Resistive Switching Devices

Transition metal oxides are promising switching layer options for RS devices. Electrochemically synthesized ZnO, TiO_2_, and CuO are transition metal oxides that are frequently used for RS memory applications, while NiO, MnO_2_, WO_3_, Ta_2_O_5_, ZrO_2_, and Co_3_O_4_ have also been reported for RS memory applications [180,181,182,183,184,185]. Nanotemplate-based synthesis has recently been reported to be a promising technique for RS devices. Recently, Kim et al. reported NiO nanowire-based devices fabricated via electrodeposition with AAO membranes. The fabricated device exhibited similar RS properties to those of NiO thin films. The authors reported that the operating voltage of the device was directly proportional to the length of the NiO nanowires [186]. In the same vein, Song et al. fabricated AAO nanotemplates based on Ni/NiO/Ni devices with cell sizes of less than 100 nm. They were able to produce NiO nanodots with a high density (1 × 10^10^/cm^2^) and uniformity. The fabricated device exhibited unipolar RS characteristics. Interestingly, NiO doping with phosphorous switched the Au/NiO/Au device from unipolar to bipolar when fabricated using AAO template-based electrodeposition. The fabricated device operated within ±1 V [187]. AAO nanotemplate-based nanopatterning techniques thus have significant potential for use in high-density memory applications [88].

Valve metal oxides and valve metal alloy oxides can also be tailored electrochemically to boost the performance of RS devices. Sophocleous et al. demonstrated a TaO_x_-based device with good memristive properties. A switching layer with a thickness of 10 nm was fabricated using electrochemical anodization, and a high ON/OFF ratio was obtained. The device was low-cost and environmentally friendly and exhibited better RS performance with possible CMOS compatibility [188]. Zaffora et al. also reported a Ta/Ta_2_O_5_/Pt-based RS device with the active switching layer fabricated using anodization in a borate buffer electrolyte. The device had a stable LRS and HRS for up to 10^6^ cycles and a retention of up to 10^4^ s. The SET and RESET times for the device were 20 ns and 165 ns, respectively [189]. In 2019, the same research group fabricated a device based on the anodization of Ta and an Al alloy for RS applications. The device had similar properties to an anodic Ta_2_O_5_-based RS device. They also fabricated a mixed Hf-Nb oxide via the anodization of Hf and Nb. Their results suggested that this electrochemical approach has significant potential for the production of high-quality thin films for RS devices [132]. 

Flexibility and transparency can expand the range of applications for RS devices. Interestingly, WO_3_-based RS devices have been reported to show both characteristics. Ji et al. fabricated a Cu/WO_3−x_/ITO-based flexible RS device using anodization. The device had a high I_ON_/I_OFF_ ratio of ~10^5^ and a stable retention of up to 5 × 10^5^ s. They also reported outstanding uniformity over 10^3^ bending cycles [190]. Qian et al. fabricated a highly transparent ITO/WO_3_/ITO RS device using cathodic electrodeposition (Figure 11a) [180]. The device had high optical transmittance (Figure 11b), a low operating voltage (Figure 11c), and a long retention time (>10^4^ s) [180]. Electrochemically deposited Mn_3_O_4_ devices can also exhibit RS characteristics. Koza et al. deposited Mn_3_O_4_ potentiostatically in an acetate electrolyte with a potential range of 0.20–0.35 V. The Mn_3_O_4_ thin film has unipolar RS properties, and the SET and RESET times of the device were 2 and 50 ns, respectively [43]. An electrodeposited δ-Bi_2_O_3_-based RS device has also been reported to be unipolar (Figure 11d) [181]. The authors concluded that the formation and rupture of Bi filaments were responsible for switching in a Pt/δ-Bi_2_O_3_/Au memory cell (Figure 11e) [181]. Recently, electrodeposited VO_2_ has also been reported for RS applications [191,192]. An electrodeposited P-type CO_3_O_4_ nanosheet-based device exhibited bipolar RS behavior and a stable RS process over 200 cycles, with data retained for up to 10^4^ s. [193]. Oxidized MoS_2_ is also an attractive material for RS applications due to its promising memristive properties. Jamilpanah et al. reported an electrodeposited MoS_2_-MoO_2_-MoO_3_-based RS device with ECM and VCM mechanisms when using Ag and Au top electrodes, respectively. Interestingly, the annealing of the composite created pure-phase MoS_2_ and MoO_3_ in Ar and air, respectively, which did not demonstrate memristive properties [194]. 

As with programmable solid-state metallization cells, liquid-state metallization cells produce good results in aqueous systems. Han et al. fabricated a device by stabilizing the aqueous growth of Cu in polycarbonate nanopores with the use of shock electrodeposition. The device exhibited considerable *I–V* characteristics and endurance for up to 500 cycles with a retention time of <10 s [195]. Polymer-based composites can also exhibit useful RS properties. For example, Xu et al. fabricated a Co-nanoparticle dispersed polypyrrole (PPy) based on an electrochemical approach for RRAM applications (Figure 11f) [182]. Metal nanoparticles were composited with a polymer, which enhanced the spontaneous oxidation of the conducting polymer. The RS mechanism of the device was based on oxygen ion mobility, which was able to be tuned by the metal nanoparticle concentration. The Cu/Co−PPy composite/Pt device exhibited stable bipolar RS (Figure 11g), which was due to oxygen ion movement within the switching layer (Figure 11h) [182]. Electrodeposited Prussian blue [196], AAO template-based electrodeposited Bi_1−x_Sb_x_ [197], and electrodeposited copper tetracyanoquinodimethan [198] have also been reported for RS memory applications. 

Table 7 summarizes the electrochemically fabricated metal oxides, composites, and polymer-based RS devices reported to date. There is a broad scope for synthesizing oxide- and polymer-based RS materials using electrochemical approaches, leading to greater RS memory performance than existing devices. Many important properties such as surface area, power consumption, switching uniformity, and reliability need to be investigated further to fabricate high-performance memory devices. Importantly, flexible and physically transient switching layers have rarely been synthesized using electrochemical approaches. In the future, these devices will play an essential role in many applications. 

## 7. Electrochemically Synthesized Resistive Switching Materials for Neuromorphic Computing Applications

In neuromorphic computing applications, electronic devices generally mimic biological synaptic properties. In biological synapses, the pre-neuron transfers an electrical/chemical signal to the post-neuron via the synaptic cleft and modulates the signal amplitude and shape of the post-synaptic spike. This process involves different biological mechanisms and is referred to as synaptic learning [199,200,201,202,203,204]. Figure 12a presents a typical biological synapse of the human brain [205]. Due to structural and mechanistic similarities, RS devices are generally used to mimic bio-synaptic properties. Artificial or electronic synaptic devices can mimic some synaptic rules and properties, including STDP, SRDP, potentiation, and depression [1]. Several electronic devices can operate as artificial synaptic devices, such as RS/memristors [92], transistors [93], and ferroelectric junctions [94]. Of these, RS/memristor devices are the most suitable candidates due to their tunable electrical properties. 

Electrochemically synthesized devices can also be employed as artificial synaptic devices, but limited research has been conducted in this area. In particular, electrochemically prepared metal oxides, chalcogenides, and polymers can mimic bio-synaptic properties. Recently, Park et al. reported electrochemically synthesized reduced graphene oxide (rGO) as a wearable textile for neuromorphic applications. They electrodeposited rGO on stainless conductive yarns following the electrochemical reduction of GO (Figure 12b,c) [205]. The fabricated device exhibited an excitatory post-synaptic current (EPSC) and paired-pulse facilitation (PPF) (Figure 12d). By controlling the amplitude of the applied pulse, the device could switch from STP to LTP. The device also exhibited stable neuromorphic properties under bending (Figure 12e,f), highlighting its suitability for integrated textile applications [205]. 

An anodic oxide is a promising option for neuromorphic applications, but only two reports of electrochemically fabricated devices containing this material are available in the literature. Chen et al. reported Pt/TiO_2−x_/Ti_-_based electronic synapse devices with a switching layer that was fabricated via anodization of a Ti substrate [206]. The electrochemical setup for anodization is depicted in Figure 12g, with the fabricated device exhibiting RS for up to 4500 switching cycles (Figure 12h). A rapid switching time (90 s) and multistate conductance were also observed for this device. PPF, paired-pulse depression, and the applied pulse waveform are presented in Figure 12i,j. The device also mimicked the potentiation and depression properties of biological synapses (Figure 12k). Choi et al. also fabricated a Pt/TaO_y_/nanoporous TaO_x_/Ta crossbar array structure via the anodization of the Ta layer in an aqueous sulfuric acid electrolyte [207]. The fabricated crossbar array structure minimized the undesired leakage current through neighboring cells. The device exhibited high non-linearity (10^4^), low synapse coupling (4.00 × 10^−5^), and good endurance and retention. The authors also mimicked basic and advanced synaptic properties such as LTP, LTD, and STDP. They reported that the device had an 89.08% recognition accuracy after 15 training epochs. As discussed earlier, very few reports are available in the literature related to electrochemically synthesized artificial synaptic devices; thus, they require further investigation. Table 8 summarizes the synaptic learning properties of electrochemically fabricated RS devices for neuromorphic computing applications.

## 8. Electrochemically Synthesized Resistive Switching Materials for Sensing Applications

Due to rapid industrialization, pollution has emerged as a major global challenge. Therefore, there is a need to identify specific pollutants and physical and biological entities in drinking water, food, and the air before they can harm human health, while bio-sensors are needed for the early diagnosis of health issues. In this context, several sensor technologies have been investigated to reduce costs, increase portability and sensitivity, and improve recycling capabilities. In particular, in recent years, many researchers have looked to fabricate sensors that utilize the RS effect, including those for the detection of ascorbic acid [75], drug levels [49], glucose/saliva [208], heavy metals [209,210], gas [211,212], and radiation [213]. For example, Figure 13a presents a memristive aptasensor that can sense drugs based on RS [49]. The change in electrical characteristics such as the voltage gap is a response to the change in the drug concentration and human serum (Figure 13b,c) [49]. Based on this, the drugs are analyzed both quantitatively and qualitatively. 

To the best of our knowledge, there have been no reports of electrochemically prepared RS materials being used for sensing applications, but the working principle of many sensors is based on electrochemical sensing. For example, a pH sensor has been developed that utilizes the RS effect at the electrolyte–oxide–semiconductor interface [214]. The device was fabricated with Pt as the bottom electrode and SiO_2_ as the switching layer. For electrochemical characterization, Pt was employed as the counter electrode and Ag/AgCl as the reference electrode. The authors reported that the threshold voltage of the device increased linearly with an increase in the pH from 1 to 12 with high linear sensitivity (46.3 mV/pH). The contact between the electrolyte and SiO_2_ layer formed an electric double layer that led to the aggregation of ions at the interface. Under forward bias, positive ions were injected into the SiO_2_ layer to form conduction filaments, and the device switched to an LRS. Under a reverse bias, the device switched to an HRS by extracting the ions from the SiO_2_ layer. The change in the concentration of positive ions in an electrolyte changes the RS properties, allowing the pH of the electrolyte to be determined [214]. Overall, electrochemically prepared materials have been employed in many sensing applications. Combining the advantages of these materials with RS holds promise for the development of novel sensor designs. 

## 9. Future Outlook 

The present literature review has shown that the electrochemical synthesis of RS devices can reliably produce high-performance memory, synaptic, and sensor devices. However, electrochemical synthesis has not been fully explored compared to other solution-processable and physical deposition techniques. Very few electrochemically fabricated devices are available, and various synthesis techniques, nano/micromaterials, and device structures need to be tested to maximize the potential of this area of research. The effects of electrochemical parameters on RS properties have also not been well explored in past studies. Therefore, future research on various electrochemically synthesized materials and corresponding engineered devices is required in the form of well-planned experiments to realize high-performance RS devices for use in memory, neuromorphic computing, sensor, and security applications.

Most previous research has employed electrochemical techniques only for the fabrication of switching layers, while standard physical deposition techniques are used for the top and bottom electrodes. However, it is possible to synthesize an entire RS device using electrochemical fabrication, which would help to reduce production costs and simplify the fabrication process. One such device has already been reported by Han et al., utilizing a bottom-up approach [168]. Working on similar lines, all-electrochemically fabricated RS devices could revolutionize non-volatile memory and neuromorphic computing. In particular, electrochemically synthesized carbon nanomaterials have significant potential for use in various fields, particularly due to their strong RS behavior. One of the advantages of electrochemical synthesis is that it allows for the uniform deposition of carbon materials on conducting substrates; thus, graphene [215], graphene oxide [216], and CNTs [217] can be used as an active switching layer and top/bottom electrodes in RS devices via electrophoretic deposition or electrodeposition. While ZnO, TiO_2_, and CuO are the most commonly electrochemically synthesized materials for RS applications, the electrochemical approach also supports the synthesis of CMOS-compatible materials such as HfO_2_, TaO_x_, and Ta_2_O_5_. Therefore, it is recommended to synthesize these materials using a variety of electrochemical synthesis techniques. 

Tuning the electrochemical parameters (such as the electrolyte precursor, electrolyte concentration, pH, deposition time, and deposition temperature) can also be used to enhance RS properties. Moreover, the morphology and/or orientation of the material can be adjusted to influence the RS properties and mechanisms [35]. Doping, metal decoration, composites, and bilayered structures have proved effective in this respect for various applications. In particular, it has been demonstrated that the decoration of metal oxides with metal [218], the use of core–shell structures [219], and the fabrication of bilayered structures can improve the RS properties of a device. Multilayered electrodeposited devices can also exhibit stronger RS properties. In addition to metal oxides, polymers [220] and metal–organic materials [221] can be deposited using electrochemical synthesis and are suitable for neuromorphic and sensing applications. However, few studies reported on electrochemically synthesized polymer-based RS devices. Therefore, further exploration of electropolymerization is recommended, as this technique has the potential to achieve impressive RS properties. In addition, PCM is an emerging RS device, and its electrochemical fabrication may pave the way for the future development of RS applications. Most commonly employed phase-change memory materials (i.e., Ge-Sb-Te) can be electrochemically deposited onto a conducting substrate in a nonaqueous electrolyte using a three-electrode system (conducting substrate: working electrode, Ag/AgCl: reference electrode, and Pt: counter electrode). Recently, Huang et al. developed GeSbTe phase-change memory cells using a non-aqueous electrodeposition technique [69]. Here, an electrolyte contained [NBu^n^_4_][GeCl_5_], [NBu^n^_4_][SbCl_4_]s, and [NBu^n^_4_]_2_[TeCl_6_]. [NBu^n^_4_]Cl was added to the CH_2_Cl_2_ as a supporting electrolyte, with Ge-Sb-Te ultimately electrodeposited at −1.75 V vs. Ag/AgCl. Similar to this approach, binary and ternary chalcogenides can be deposited on conductive substrates for RS applications. In addition, given that most top and bottom electrodes are either metal or conducting materials, it is possible to exploit electrodeposition to fabricate and/or pattern top and bottom electrodes. Considering the previously identified research gaps in the literature, we believe that the electrochemical approach holds significant promise for the fabrication of state-of-the-art RS devices. 

## 10. Conclusions

This review outlined recent advances in electrochemically synthesized memory, synaptic, and sensor devices, with particular emphasis on their established RS mechanisms and the major challenges associated with their manufacturing processes, including material matching, electrode caps, and layers of insulator components. Research and development in recent years have focused on revolutionizing metal-oxide materials, which have been used as core components of conventional metal–insulator–metal memristors. The possible applications, prospects, and future development of various materials and their synthesis routes were outlined in this paper. Industrial manufacturing is necessary for the commercialization of RS devices, and electrochemical approaches have several features that make them commercially viable, including their ease of fabrication and low-temperature processing environment. The precise control of the thickness of the switching layer and the composition of materials by tuning the electrochemical parameters can lead to better switching voltages, endurance, retention, and switching times. The bottom-up synthesis of entire RS devices using electrochemical techniques can also be employed to develop CMOS-compatible memory, synaptic, and sensor devices. In addition to this, RS devices can be miniaturized using template-based electrodeposition techniques. Oxides, polymers, organic materials, and composite-based devices can be fabricated using electrochemical synthesis techniques. Therefore, this research area needs to be explored further for the production of high-performance RS devices that can be employed in various applications. In particular, an understanding of RS and charge transport mechanisms will help to optimize future RS devices, so new models and simulation techniques will be needed in the coming years. Finally, device design and the engineering of the device structure are some issues that require close attention. If these research directions are followed, multifunctional RS devices have the potential to be the future of electronics.

## Figures and Tables

**Figure 1 nanomaterials-13-01879-f001:**
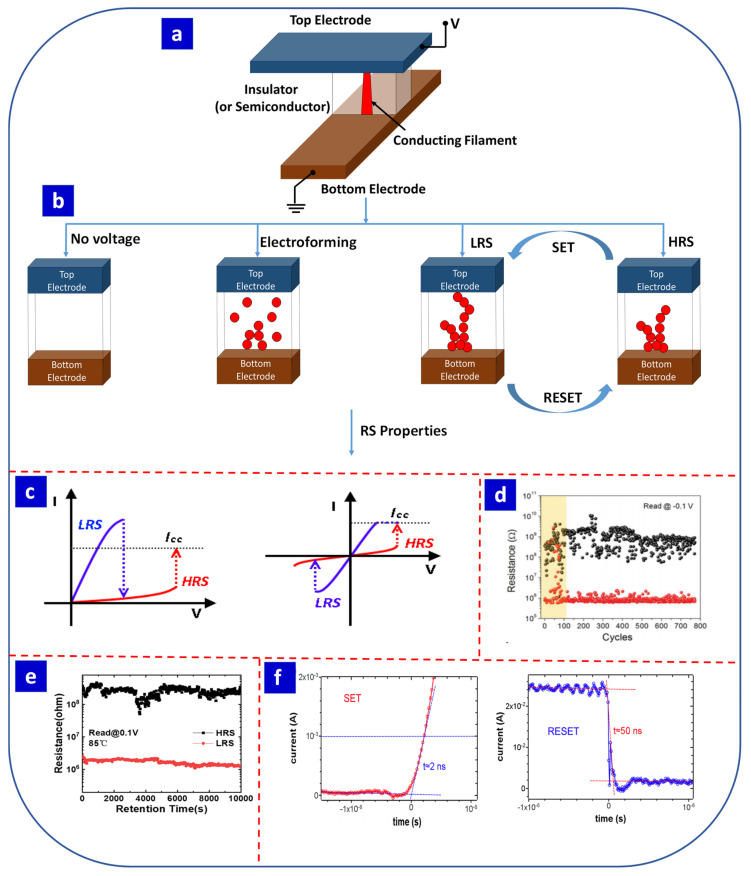
(**a**) Schematic of a metal–insulator–metal structure. Redrawn from [39]. (**b**) Schematic of the resistive switching (RS) effect based on filamentary conduction within a virgin cell and the electroforming, SET, and RESET processes. Redrawn from [40]. (**c**) Unipolar and bipolar RS effects. Reprinted with permission from [41]. Copyright © 2023, MDPI. (**d**) The endurance of a device measured at a read voltage of −0.1 V. Reprinted with permission from [3]. Copyright © 2023, Wiley-VCH. (**e**) Retention characteristics of an RS device at 85 °C. Reprinted with permission from [42]. Copyright © 2023, Springer. (**f**) Current-time transient measurements during the SET and RESET processes. Reprinted with permission from [43]. Copyright © 2023, American Chemical Society.

**Figure 4 nanomaterials-13-01879-f004:**
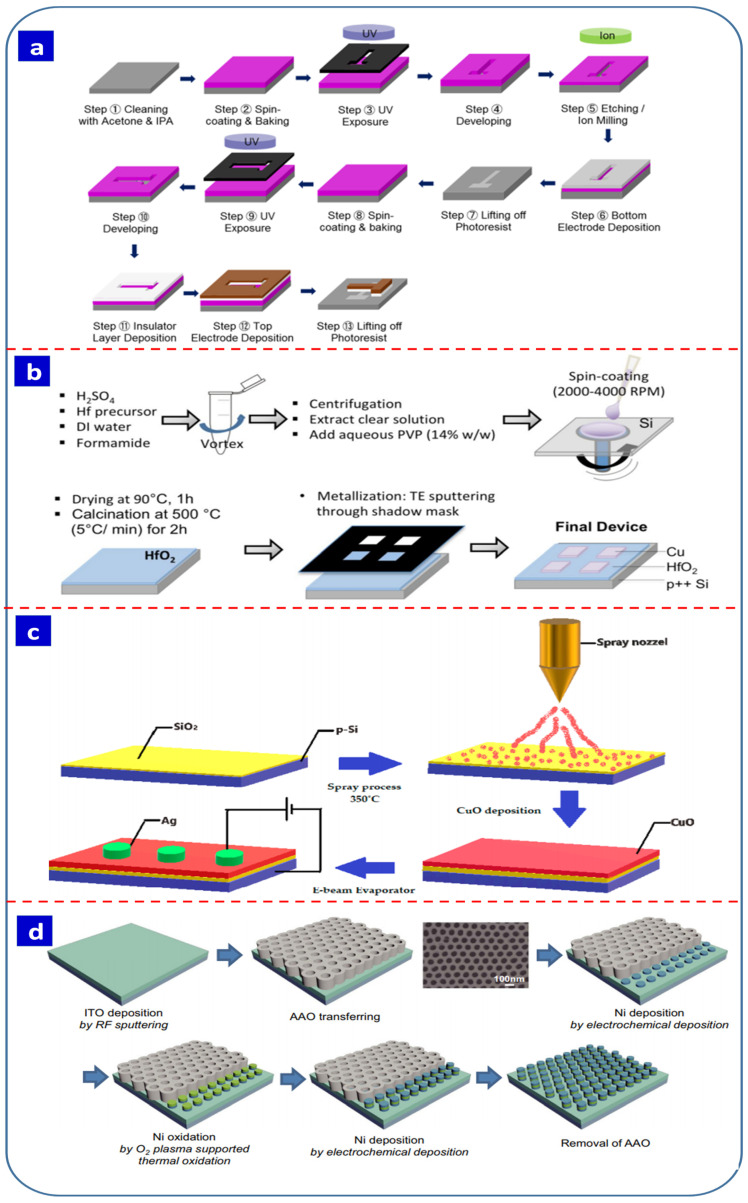
(**a**) Fabrication process for a Pt/HfO_2_/Ti RS device, including lithography, etching, and deposition. Reprinted with permission from [100]. Copyright © 2023, American Chemical Society. (**b**) Steps involved in spin-coated Cu/HfO_2_/p^++^ Si RS devices. Reprinted with permission from [27]. Copyright © 2023, Springer-Nature. (**c**) Steps involved in Ag/CuO/SiO_2_/p-Si memory device fabrication. Reprinted with permission from [101]. Copyright © 2023, MDPI. (**d**) Process flow for nanoscale Ni/NiO/Ni ReRAM device fabrication using electrochemical deposition with AAO as a template mask. Reprinted with permission from [102]. Copyright © 2023, Springer-Nature.

**Figure 6 nanomaterials-13-01879-f006:**
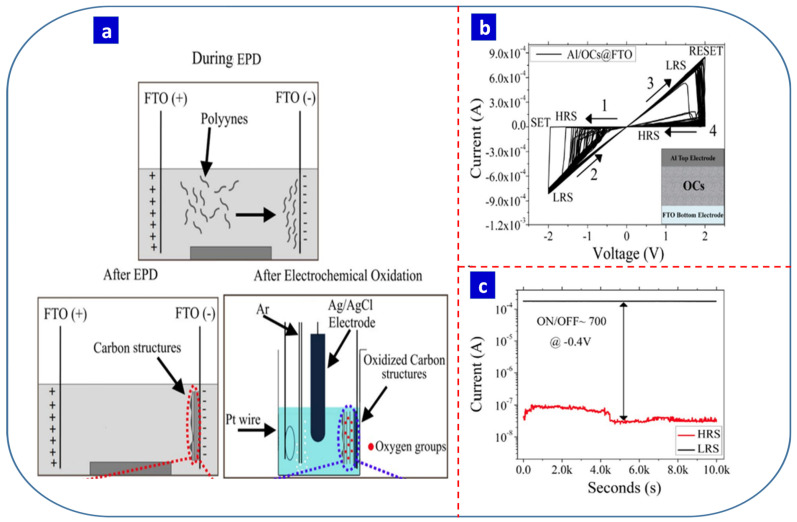
(**a**) Steps involved in the synthesis of carbon structures, such as the deposition of polyynes during electrophoretic deposition, after electrophoretic deposition, and after electrochemical oxidation. (**b**) Cyclic stability and (**c**) retention properties of an Al/OCs@FTO device. Reprinted with permission from [137]. Copyright © 2023, Springer-Nature.

**Figure 7 nanomaterials-13-01879-f007:**
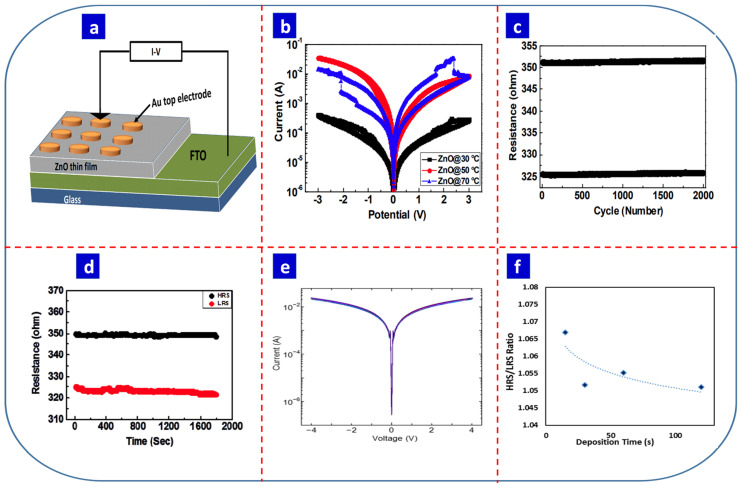
(**a**) *I–V* measurement setup for electrochemically deposited ZnO thin films using an electrochemical workstation. (**b**) *I–V* curves for ZnO thin films prepared at various temperatures using an electrochemical method on a semi-log scale. (**c**) Endurance and (**d**) results for a ZnO nanorod-based memory device. Reprinted with permission from [36]. Copyright © 2023, AIMS Press. (**e**) Semi-logarithmic *I–V* curve for electrochemically synthesized ZnO thin films. (**f**) Deposition-time-dependent R_OFF_/R_ON_ ratio for a Au/ZnO/ITO memristor. Reprinted with permission from [148]. Copyright © 2023, IOP.

**Figure 8 nanomaterials-13-01879-f008:**
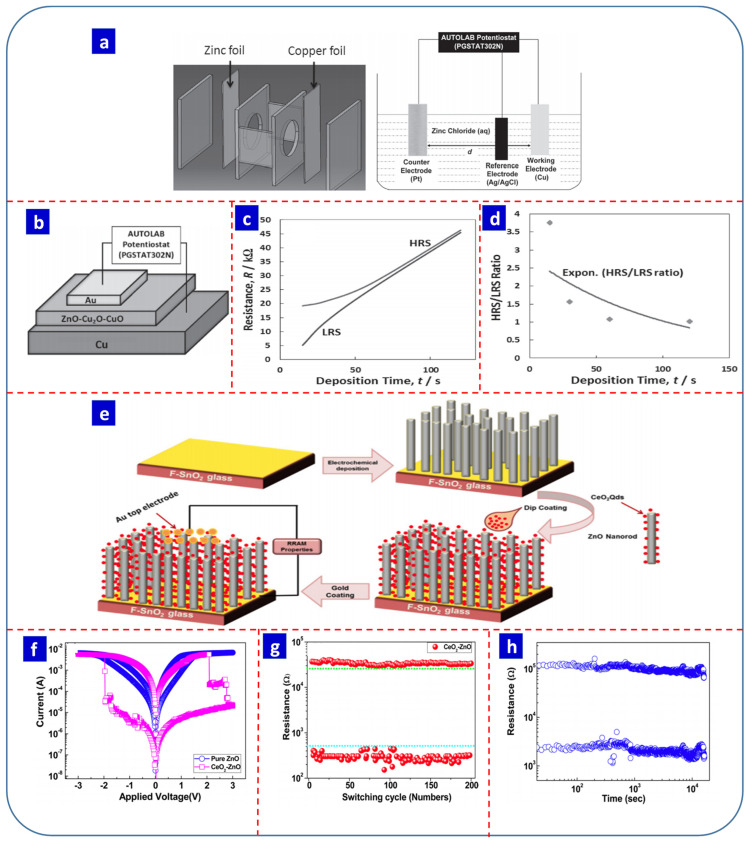
(**a**) Schematic of an electrolytic cell holder assembly and diagrammatic representation of a typical electrodeposition setup for a three-electrode system. (**b**) Schematic of a Au/ZnO-Cu_2_O-CuO/Cu device for *I–V* measurements. (**c**) Change in the resistance from an HRS to an LRS with a change in the electrodeposition time. (**d**) HRS/LRS ratio at different electrodeposition times. Reprinted with permission from [153]. Copyright © 2023, Japan Institute of Metals and Materials. (**e**) Schematic representation of the steps involved in the electrochemical synthesis of a CeO_2_–ZnO nanocomposite and the fabrication process for a memory device. (**f**) *I–V*, (**g**) endurance, and (**h**) retention characteristics for a Au/CeO_2_−ZnO/FTO device. Reprinted with permission from [155]. Copyright © 2023, American Chemical Society.

**Figure 9 nanomaterials-13-01879-f009:**
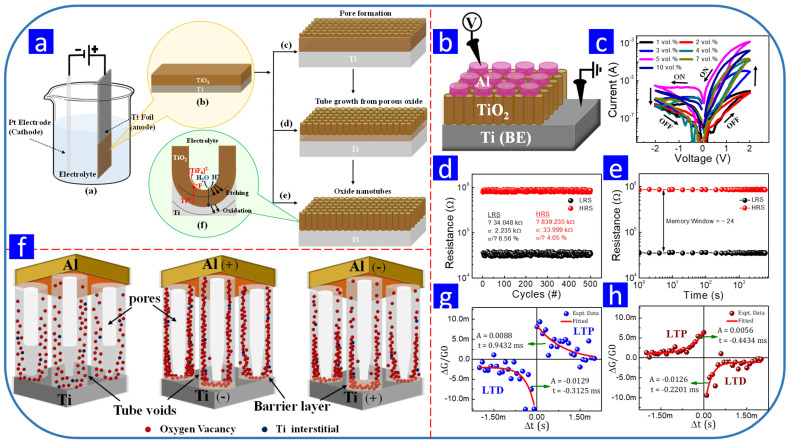
(**a**) Schematic representation of TiO_2_ nanotube formation via electrochemical anodization. (**b**) Device structure, (**c**) *I–V* characteristics, (**d**) endurance, and (**e**) retention for an Al/TiO_2_/Ti memory device. (**f**) Schematic illustration of the filamentary RS effect in the nanotube walls. (**g**,**h**) STDP-based synaptic learning rules, e.g., antisymmetric Hebbian and antisymmetric anti-Hebbian. Reprinted with permission from [133]. Copyright © 2023, American Chemical Society.

**Figure 10 nanomaterials-13-01879-f010:**
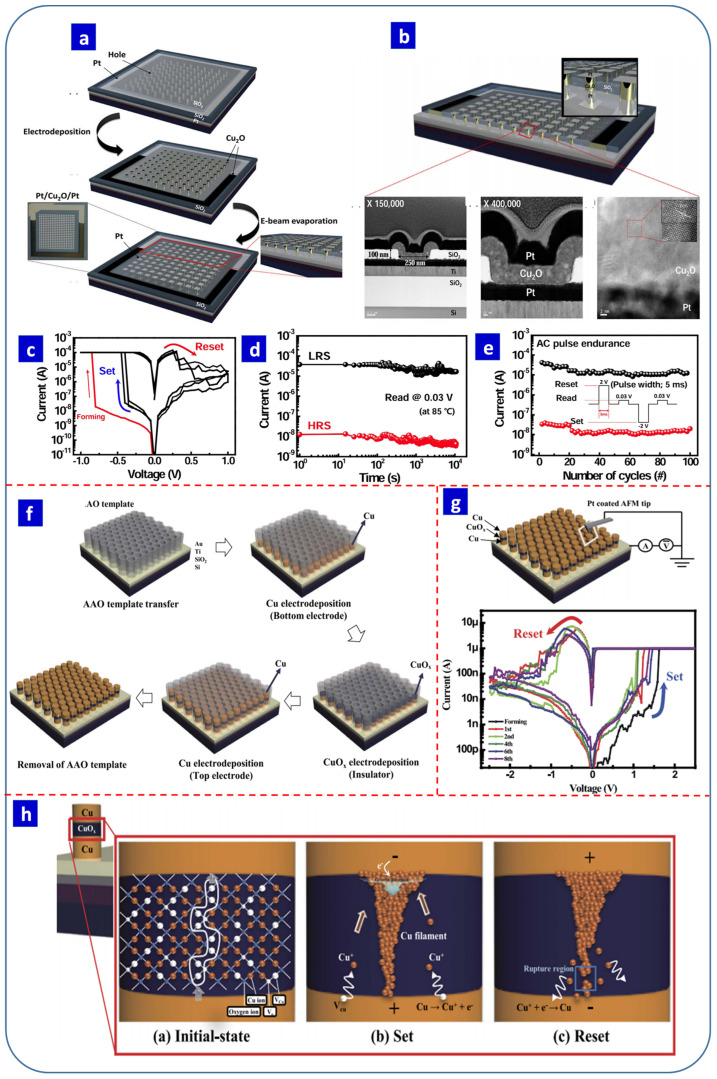
(**a**) Fabrication process for an electrochemically deposited Cu_2_O-based memory device with a 250 nm cell size and holes patterned in the wafer. (**b**) Schematic representation of an electrochemically fabricated Pt/Cu_2_O/Pt RS device and cross-sectional TEM images at different magnifications. (**c**) *I–V*, (**d**) retention, and (**e**) endurance characteristics for Pt/Cu_2_O/Pt. Reprinted with permission from [167]. Copyright © 2023 Springer-Nature. (**f**) The fabrication process for nanoscale Cu/CuO_x_/Cu RS memory using AAO nanotemplate-based electrodeposition. (**g**) Schematic setup for the electrical characterization of Cu/CuO_x_/Cu nanodot memory devices using conductive atomic force microscopy and the *I–V* characteristics of nanoscale Cu/CuO_x_/Cu memory. (**h**) RS mechanism for a Cu/CuO_x_/Cu device, including the initial, SET, and RESET states. Reprinted with permission from [168]. Copyright © 2023, Springer-Nature.

**Figure 11 nanomaterials-13-01879-f011:**
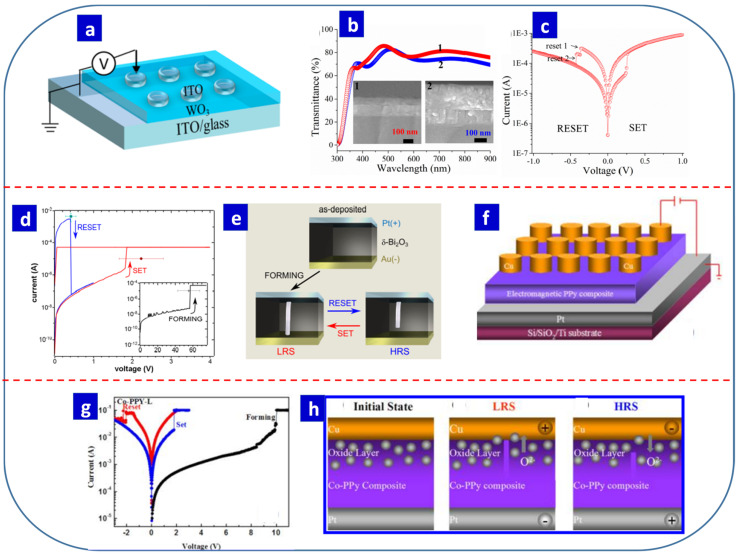
(**a**) Schematic view of an electrochemically fabricated ITO/WO_3_/ITO transparent device array for electrical characterization. (**b**) Optical transmittance spectra for ITO/WO_3_/ITO/glass devices with different thicknesses of WO_3_. (**c**) *I–V* characteristics of an electrodeposited (∼50 nm) WO_3_-based memory cell. Reprinted with permission from [180]. Copyright © 2023, American Chemical Society. (**d**) *I–V* curves for a Pt/δ-Bi_2_O_3_/Au cell during the FORMING, SET, and RESET steps at a rate of 0.7 V/s. (**e**) Schematic illustration of the electroforming, formation, and rupture of a conducting filament during the SET and RESET steps in a Pt/δ-Bi_2_O_3_/Au device. Reprinted with permission from [181]. Copyright © 2023, American Chemical Society. (**f**) Electrochemically deposited Co−PPy-composite memory device structure. (**g**) *I–V* characteristics for a bipolar RS Co−PPy-L sample. (**h**) Schematic representation of the RS mechanism during the SET and RESET processes in a Co−PPy-composite device. Reprinted with permission from [182]. Copyright © 2023, American Chemical Society.

**Figure 12 nanomaterials-13-01879-f012:**
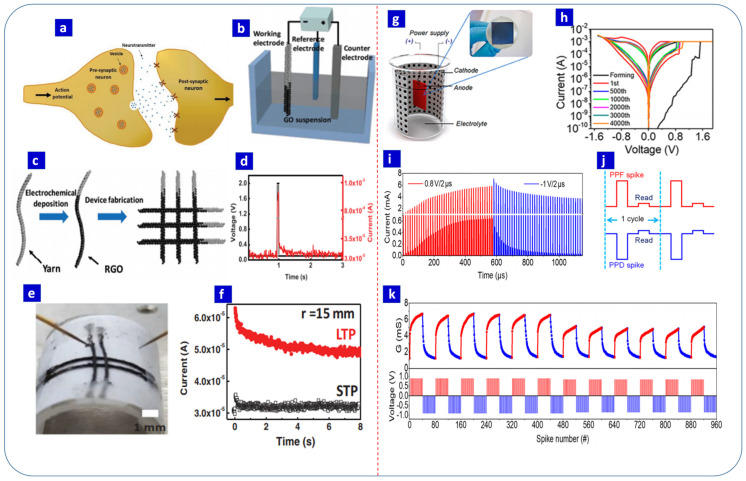
(**a**) Schematic of a biological synapse. (**b**) Schematic representation of the electrochemical setup for the electrodeposition of reduced graphene oxide (rGO) on yarn. (**c**) Fabrication of an electrodeposited rGO-based yarn device. (**d**) EPSC characteristics of rGO-based yarn with pulse stimulation (2 V, 50 ms). (**e**) Image of an rGO-based yarn device during bending measurements. (**f**) STP and LTP characteristics of the bent device after 10 consecutive pulses. Reprinted with permission from [199]. Copyright © 2023, Wiley-VCH. (**g**) Schematic view of an electrochemically prepared TiO_2−x_ thin film on a Ti substrate. (**h**) *I−V* characteristics of a Pt/10 nm TiO_2−x_/Ti device. (**i**) Paired-pulse facilitation and paired-pulse depression of an anodized memristive synapse and (**j**) its applied pulse waveform. (**k**) Potentiation and depression characteristics of anodic TiO_2−x_ synapses. Reprinted with permission from [200]. Copyright © 2023, American Chemical Society.

**Figure 13 nanomaterials-13-01879-f013:**
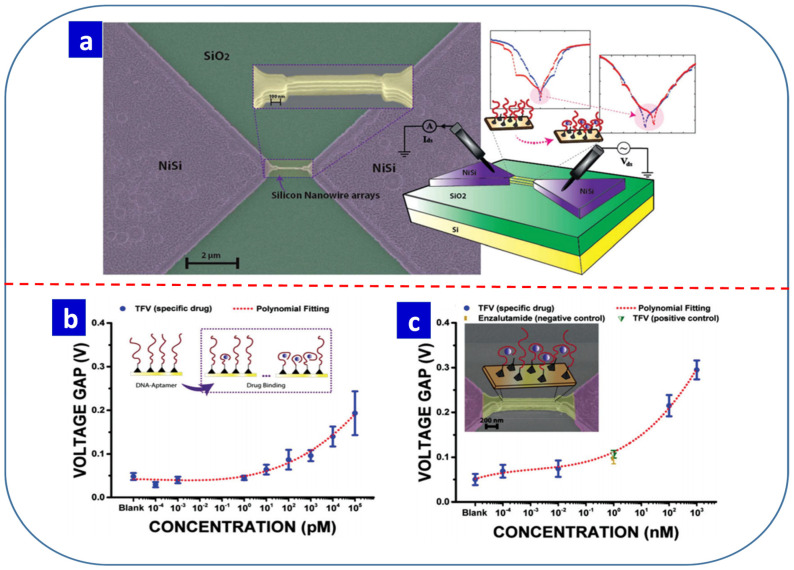
(**a**) Schematic illustration of a memristive aptasensor and its semi-logarithmic *I–V* characteristics. (**b**) Detection of a drug through electrical hysteresis variation in a buffer and (**c**) human serum. Reprinted with permission from [49]. Copyright © 2023, Royal Society of Chemistry.

**Table 2 nanomaterials-13-01879-t002:** The advantages and disadvantages of switching layer synthesis techniques.

Sr. No.	Method	Advantages	Disadvantages	Copper Oxide-Based Device Performance	Refs.
1	Sputtering	High-quality uniform switching layer with good adhesion Large-area coating possible Thickness can control up to a few nm Low-melting-point substrate can be used for deposition CMOS compatible	Expensive method Substrate damage risk due to continuous ion bombardment Vacuum environment required No morphological control	V_SET_: 0.7 V V_RESET_: −0.5 V Endurance: 10^3^ cycles Retention: 10^4^ s Memory Window: 10^5^	[105,106,116]
2	Chemical vapor deposition	Uniform and reproducible switching layer Good adhesion of switching layer CMOS compatible	Expensive method High reaction time Low deposition rate Presence of corrosive gases High-temperature synthesis	V_SET_: −2 V V_RESET_: 1.5 V Endurance: 10 cycles Retention: 50 s Memory Window: 7	[117,118]
3	Spray pyrolysis	Operates in atmospheric pressure Large-surface-area deposition can be possible More durable coating than vacuum deposition techniques Multilayer deposition can be possible	Quite expensive High-temperature requirement	V_SET_: 1.5 V V_RESET_: −0.65 V Endurance: 50 cycles Retention: 10^3^ s Memory Window: 10^3^	[110,116]
4	Spin coating	Easy mode of operation Cost-effective technique The substrate can be an insulator or conductor Fast deposition rate	Difficult to control uniform thickness and morphology Not applicable for large-scale deposition	V_SET_: 2.25 V V_RESET_: −2.25 V Endurance: 10^2^ cycles Retention: 800 s Memory Window: 50	[109,119]
5	Hydrothermal	Easy mode of operation Ability to grow high-quality switching layer Easy control of morphology Cost-effective synthesis	High-temperature fabrication method Time-consuming method In situ observation of crystal growth is not possible Not applicable for large surface growth	V_SET_: 1 V V_RESET_: −1 V Endurance:10 cycles Retention: 600 s Memory Window: 10^2^	[111,116,120]
6	Sol–gel	Simple technique Low-cost and reproducible method Suitable for various substrates Large-area coating possible	The deposition period is long High-temperature operation for phase conversion A thick deposition is not possible Difficult to form porous films	-	[116,118,120]
7	Electrochemical	Relatively very simple and cheap Large-area coating possible with uniform thickness Fast deposition rate Low-temperate synthesis Morphology can be controlled by tuning the deposition parameters Control over thickness and deposition time Good adhesion and durability of the switching layer Multilayer deposition can be possible Eco-friendly approach Industrial applicability The whole MIM structure can be fabricated	Relatively quite high energy requirement Deposition on conducting substrate only	V_SET_: 0.7 V V_RESET_: −0.7 V Endurance: 5 × 10^3^ cycles Retention: 10^3^ s Memory Window: 10^2^	[104,116,118,120]

**Table 3 nanomaterials-13-01879-t003:** Synthesis parameters and properties of the electrochemically synthesized carbon material-based RS devices.

RS Material	Synthesis Method	Electrolyte	Applied Voltage	Time	V_SET_ (V)	V_RESET_ (V)	ON/OFF Ratio	Endurance Cycles	Data Retention (s)	Mode	Device Structure	Application	Ref.
Carbon nanowalls	Electrophoretic	Polyyne solution by arc discharge	30 V	1 h	−2	2	-	150	5 × 10^4^	Bipolar	Al/CNWs/FTO	Memory Storage	[35]
Carbon structures	Electrophoretic	Polyyne solution by arc discharge	30 V	2 h	−2	2	700	50	1 × 10^4^	Bipolar	Al/OCs@FTO	Memory Storage	[137]
Graphene oxide	Electrophoretic	0.2 mol/L GO solution	10 V/cm	1 min	1.7 V	−1.7 V	10	-	10^2^	Bipolar	Al/GO/ITO	Memory Storage	[147]

**Table 4 nanomaterials-13-01879-t004:** Synthesis parameters and properties of the electrochemically synthesized ZnO-based RS devices.

RS Material	Synthesis Method	Electrolyte	Applied Voltage/Current	Temp. °C	Time	V_SET_ (V)	V_RESET_ (V)	ON/OFF Ratio	Endurance Cycles	Data Retention (s)	Mode	Device Structure	Application	Ref.
ZnO	Electrodeposition	0.01 M Zn(NO_3_)_2_·6H_2_O	1 mA	70	20 min	3	−3	-	2000	1800	Bipolar	Au/ZnO rod/FTO	Memory storage	[36]
ZnO	Electrodeposition	0.01 M ZnCl_2_	1.5 V	-	20 min	1.9	−2	-	-	-	Bipolar	-	Memory storage	[149]
ZnO	Cyclic voltammetry deposition	0.01 M Zn(NO_3_)_2_·6H_2_O, 0.1 M KNO_3_	±0.9 V	65 ± 2	300 s	1	−1	50	200	10^4^	Bipolar	Pt/ZnO/ITO	Memory storage	[104]
ZnO	Electrodeposition	5 mM Zn(NO_3_)_2_·6H_2_O and HNO_3_	−0.9 V	80	1000 s	2.7	−2.5	-	100	1000	Bipolar	Au/ZnO/AZO	Memory storage	[150]
ZnO	Electrodeposition	0.005M ZnCl_2_	2.0 V	RT	-	4	−4	-	100	-	Bipolar	Au/ZnO/ITO/PET	Memory storage	[148]
ZnO	Electrodeposition and hydrothermal	ZnSO_4_. 7H_2_O Na_2_SO_4_ H_3_BO_3_, H₂SO₄	−1.5 V	25	400 s	−4	5	-	-	10	Bipolar	Pt (AFM tip)/ZnO/Zn	Memory storage	[151]
Ti-ZnO	Electrodeposition	0.1 M Zn (NO_3_)_2_·6H_2_O with 2% (NH_4_)_2_TiF_6_	1 mA	75	30 min	2.7	−2.9	14	200	2000	Bipolar	Au/Ti-ZnO/ITO	Memory storage	[152]
ZnO-Cu_2_O-CuO	Electrodeposition	0.005 M ZnCl_2_	2.0 V	RT	120 s	2	−2	-	-	-	Bipolar	Au/ZnO-Cu_2_O-CuO/Cu	Memory storage	[153]
ZnO-Cu_2_O-CuO	Electrodeposition	0.005 M ZnCl_2_	1 V	-	-	2	−2	1.08	-	-	Bipolar	Au/ZnO-Cu_2_O-CuO/Cu	Memory storage	[154]
CeO_2_−ZnO	Electrodeposition and dip coating	0.01 M Zn(NO_3_)_2_∙6H_2_O	2 mA	75	30 min	−1.9	2.08	-	200	10,000	Bipolar	Au/CeO_2_−ZnO/FTO	Memory storage	[155]
ZnO-Si	Anodization and sol–gel	-	-	-	-	10	−10	~2.4	100	-	Bipolar	Al/ZnO/Si/Al	Memory storage	[156]

**Table 5 nanomaterials-13-01879-t005:** Synthesis parameters and properties of the electrochemically synthesized TiO_2_-based RS devices.

RS Material	Synthesis Method	Electrolyte	Applied Voltage	Temp. °C	Time	V_SET_ (V)	V_RESET_ (V)	ON/OFF Ratio	Endurance Cycles	Data Retention (s)	Mode	Device Structure	Application	Ref.
TiO_2_	Anodization	0.27 M NH_4_F,glycerol	30 V	-	10 s	1	−1	-	3	-	Bipolar	Ag/TiO_2_/Ti	Memory Storage	[37]
TiO_2_	Electrodeposition	(NH_4_)_2_TiF_6_ and HBO_3_	−0.8 V	90	30 min	2.5	−1.3	-	7000	-	Bipolar	Au/TiO_2_/FTO	Memory Storage	[157]
TiO_2_	Anodization	1 M H_2_SO_4_	10 V	-	15 min	0.24	−0.24	25	100	-	Bipolar	Au/TiO_2_/Ti	Memory Storage	[158]
TiO_2_	Anodization	HF and H_3_PO_4_	-	-	2 h	−1	1	-	20	-	Bipolar	Pt/TiO_2_/Ti	Memory Storage	[159]
TiO_2_	Anodization	0.1 mol/L H_2_SO_4_	20 V	-	30 min	2	−2	-	50	10^3^	Bipolar	Pt/TiO_2_/Ti	Memory Storage	[160]
TiO_x_	CV Anodization	100 mM CH_3_SO_3_H, Mn(CH _3_SO_3_)_2_	−0.5 to 10 V	-	-	1	−1		600	-	Bipolar	Ag/TiOx/Ti/Au	Memory Storage	[164]
TiO_2_	Anodization	1 M H_3_PO_4_	10 V	-	20 min	−2	2	-	20		Bipolar	Cu/TiO_2_/Ti	Memory Storage	[165]
TiO_2_	Anodization	0.5 wt% NH_4_F,ethylene glycol, water	60 V	RT	30 s	2	−2	-	20	-	Bipolar	Ti/TiO_2_/PAA/Pt	Memory Storage	[166]
TiO_2_	Anodization	0.25 wt% NH_4_F and ethylene glycol	60 V	-	5 min	0.37	−0.4	25	16	-	Bipolar	Ag/TiO_2_/Ti	Memory Storage	[161]
Core–shell copper nanowire-TiO_2_ nanotube	Anodization	0.5 wt% NH_4_F, 3 vol% H_3_PO_4_ 3 vol% DI water in ethylene glycol	80 V	RT	2 h	1.8	−2.5	40	10,000	10^4^	Bipolar	Au/TiO_2_-Cu/Ti	Memory Storage	[163]

**Table 6 nanomaterials-13-01879-t006:** Synthesis parameters and properties of the electrochemically synthesized copper oxide-based RS devices.

RS Material	Synthesis Method	Electrolyte	Applied Voltage/Current Density	pH	Time	V_SET_ (V)	V_RESET_ (V)	Forming Voltage	ON/ OFF Ratio	Endurance Cycles	Data Retention (s)	Mode	Device Structure	Application	Ref.
Cu_2_O	Electrodeposition	0.2 M CuSO_4_.5H_2_O, 0.2 M L(+)-tartaric acid, 3 M NaOH	1 mA/cm^2^	-	-	4	1	34 ± 6.7 V	10,000	170	8 × 10^4^	Unipolar	Au−Pd/Cu_2_O/Au	Memory Storage	[175]
Cu_2_O	Electrodeposition	0.6 M CuSO_4_·5H_2_O, 3 M lactic acid, 2M NaOH	1 mA/cm^2^	9	-	−0.36	0.31	~0.8	>1000	100	10^4^	Bipolar	Pt/Cu_2_O/Pt	Memory Storage	[167]
Cu_2_O	Electrodeposition	0.6 M CuSO_4_·5H_2_O, 3 M lactic acid, 2 M NaOH	1 mA/cm^2^	9	30 s	1.3	−0.75	1.5	~1000	8	-	Bipolar	Cu/CuO_x_/Cu	Memory Storage	[168]
Cu_2_O	Electrodeposition	1 M CuSO_4_.5H_2_O, 6 M sulfuric acid, NaOH	1–4 V	-	-	~−2	2	5	-	-	-	Bipolar	Al/Cu_2_O/FTO	Memory Storage	[176]
CuO_x_	Electrodeposition	0.6 M CuSO_4_.5H_2_O, 3 M lactic acid, NaOH	4 mA	-	30 s	0.5	−0.3	0.75	-	100	10^4^	Bipolar	Ni/CuOx/Ni	Memory Storage	[177]
CuO_x_	Electrodeposition	0.6 M CuSO_4_.5H_2_O, 3 M lactic acid, NaOH	−0.4 V vs. Ag/AgCl	9 and 11.5	100 s	2.5	−3	4	-	100	10^4^	Bipolar	Pt/CuOx/CuOx/Pt	Memory Storage	[178]

**Table 7 nanomaterials-13-01879-t007:** Synthesis parameters and properties of different electrochemically synthesized RS devices.

RS Material	Synthesis Method	Electrolyte	Applied Voltage/Current Density	pH	Time	V_SET_ (V)	V_RESET_ (V)	ON/OFF Ratio	Endurance Cycles	Data Retention (s)	Mode	Device Structure	Application	Ref.
NiO	Electrodeposition and thermal oxidation	Ni_2_SO_4_ ·6H_2_O (0.5M) H_3_PO_3_ (0.67 M)	3 mA/cm^2^	-	-	1.2	0.52	-	100	-	Unipolar	W/NiO/Au	Memory Storage	[186]
NiO	Electrodeposition and thermal oxidation	Ni(H_2_NSO_3_)_2_, NiCl_2_·6H_2_O, H_3_BO_3_, CH_3_COONa, H_2_SO_4_	4 mA/cm^2^	3.4	1 min	9	6	~100	10	-	Unipolar	Ni/NiO/Ni	Memory Storage	[102]
P-doped NiO	Electrodeposition and thermal oxidation	1 M Ni_2_SO_4_ ·6H_2_O, 0.2 M NiCl₂, and 0.5 M H_3_BO_3_	5 mA/cm^2^	-	5 min	0.8	−0.7	-	-	-	Bipolar	Au/P-doped NiO/Au	Memory Storage	[187]
TaOx	Anodization	1 mM citric acid	-	-	-	4.5	−5	80	-		Bipolar	Ta/TaOx/Pt	Memory Storage	[188]
Ta_2_O_5_	Anodization	H_3_BO_3_, Na_2_B_4_O_7_	-	8	-	0.45–0.65 V	0.45 0.65 V	>10	10^6^	10^4^	Bipolar	Ta/Ta_2_O_5_/Pt	Memory Storage	[189]
Ta_2_O_5_	Anodization	0.42 M H_3_BO_3_, 0.08 M Na_2_B_4_O_7_	10 mV s^−1^	8	-	0.45	−0.9	-	1000	10^4^	Bipolar	Ta/anodic Ta_2_O_5_/Pt	Memory Storage	[132]
WO_3−x_	Anodization	NH_4_F, ethylene glycol	20 V	-	20 s	1	−1	10^5^	1000	10^5^	Bipolar	Cu/WO_3−x_/ITO/PET	Memory Storage	[190]
WO_3_	Electrodeposition	Na_2_WO_4_·H_2_O, H_2_O_2_, HClO_4_	−0.7 V	1.2	60 s	0.25	−0.42	-	320	>10^4^	Bipolar	ITO/WO_3_/ITO	Memory Storage	[180]
Mn_3_O_4_	Electrodeposition	Mn(II) acetate, sodium acetate	0.3 V	6	-	0.72	2.5	-	30	-	Unipolar	AuPd/Mn_3_O_4_/Au	Memory Storage	[43]
δ-Bi_2_O_3_	Electrodeposition	0.1 M Bi(NO_3_)_3_, 0.25 M L-tartaric acid and 2.5 M KOH	5 mA/cm^2^	-	-	2.2	0.4	10^6^	100	5 × 10^4^	Unipolar	Pt/δ-Bi_2_O_3_/Au	Memory Storage	[181]
VO_2_	Electrodeposition	-	-	-	-	0.5	−0.5	-	-	-	Bipolar	Ag/VO_2_/Pt	Memory Storage	[191]
CO_3_O_4_	Electrodeposition	0.1 M Co(NO_3_)_2_·6H_2_O	−0.8 V	-	20 min	1.5	−1.93	-	200	>10^4^	Bipolar	Au/CO_3_O_4_/ITO	Memory Storage	[193]
MoS_2_-MoO_2_-MoO_3_	Electrodeposition	0.39 M Na_2_MoO_4_·2H_2_O and 0.88 M Na_2_S·5H_2_O	-	7	-	1.15 0.2	−0.5 −0.1	-	-	-	Bipolar	Al/MoSO/FTO Ag/MoSO/FTO	Memory Storage	[194]
Prussian blue	Electrodeposition	0.25 mM K_3_Fe(CN)6, 0.25 mM FeCl_3_, 1.0 M KCl and 5.0 mM HCl	0.3 V	2	-	−0.6 V	~+0.6 V	-	200	-	Bipolar	Au/Prussian blue/Ag	Memory Storage	[196]
Bi_1−x_Sb_x_	Electrodeposition	0.05 M SbCl_2_ 0.05 M Bi(NO_3_)_3_·5H_2_O dimethyl sulfoxide	−0.8 V	-	-	1.35	−1.2 V	10^4^	300	-	Bipolar	Pt/Bi_1-x_Sb_x_/Pt	Memory Storage	[197]
Co−PPy composite	Electrodeposition (CV)	0.1 mol/L Na_2_SO_4_, 0.001 mol/L sodium dodecylbenzenseulfonate, and 0.1 mol/L pyrrole Co nanoparticles	0.5–1.0 V	3	-	1.07 to 2.3	−1.29 to −2.3	-	-	10^4^	Bipolar	Cu/Co−PPy composite/Pt	Memory Storage	[182]

**Table 8 nanomaterials-13-01879-t008:** Synthesis parameters and synaptic learning properties of the electrochemically synthesized RS devices.

RS Material	Synthesis Method	Electrolyte	Deposition Potential/ Current Density	Temp.	V_SET_ (V)	V_RESET_ (V)	Device Structure	Cross Point Array	Synaptic Properties	Advantages	Ref.
TiO_2_ Nanotubes	Anodization	Ethylene glycol + NH_4_F + distilled water	30 V	-	2	−2	Al/TiO_2_ nanotubes/Ti	-	Potentiation–depression, STDP-based Hebbian learning rules	Low-cost artificial synaptic device that can mimic the basic and advanced synaptic learning properties	[133]
Cu_X_O	Anodization	3 M KOH	10 mA/cm^2^ for 150 s	-	8	−8	Pt/Cu_X_O/Cu	-	Potentiation–depression, STDP-based Hebbian learning rules	Low-cost artificial synaptic device that can mimic the basic and advanced synaptic learning properties	[169]
Reduced Graphene Oxide	Electrophoretic	Aqueous suspension of graphene oxide	−1.1 V	60 °C	-	-	Conductive yarns/RGO/conductive yarns	2 × 2	EPSC, PPF, and a transition from short-term plasticity to long-term plasticity	Operated stably without degradation during mechanical bending; artificial synapses for wearable neuromorphic systems	[205]
Anodic TiO_2−x_	Anodization	0.5 M NaOH	2.5 V	-	0.8	−1.6	Pt/anodic TiO_2−x_/Ti	-	Potentiation and depression	A cheap and effective route to fabricate competitive electronic synapses	[206]
TaO_y_/nano-porous TaO_x_	Anodization	Aqueous H_2_SO_4_ and HF	50 V for 10 s	-	4	−4	Pt/TaO_y_/nanoporous TaO_x_/Ta	4 × 4	LTP, LTD, and STDP	89.08% recognition accuracy for MNIST handwritten digit images	[207]

## Data Availability

The data presented in this study are available on request from the corresponding author.

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
