# Peer review of "Review of Electrochemically Synthesized Resistive Switching Devices: Memory Storage, Neuromorphic Computing, and Sensing Applications"

_nanomaterials, 2023, doi:10.3390/nano13121879_

Round 1

Reviewer 1 Report

Good.

Reviewer 3 Report

The paper proposes a review of electrochemically synthesized resistive switching devices. A large spectrum of application is covered: memory storage, neuromorphic and sensing devices.  A focus on the fabrication techniques employed for Resistive Switching (RS) devices is also provided.

The way The RS device applications are addressed is too superficial. For instance, in Table 1, requirements for memory storage, neuromorphic and sensing are almost identical, which didn't reflect reflects the current state-of-the-art. 3D stacking is missing in the memory storage part, neuromorphic computing only addresses spiking neural networks, etc.

Although the technical aspect related to the fabrication techniques for resistive switching devices is well covered, the scientific output of the proposition is not clear as well as the final result of the work regarding the state-of-the-art. 

Authors should mention in the introduction what problem to solve, what new information is provided regarding the state-of-the-art. Where is this leading, and what are the next steps? 

Applications of each RS stack should be linked to Table 5, 6 and 7 presented materials/materials stack presented to bring more credit to the contribution. 

NA
